# MODEL SWARMS: COLLABORATIVE SEARCH TO ADAPT LLM EXPERTS VIA SWARM INTELLIGENCE

## ABSTRACT

We propose MODEL SWARMS, a collaborative search algorithm to adapt LLMs via *swarm intelligence*, the collective behavior guiding individual systems. Specifically, MODEL SWARMS starts with a pool of LLM experts and a utility function. Guided by the best-found checkpoints across models, diverse LLM experts collaboratively move in the weight space and optimize a utility function representing model adaptation objectives. Compared to existing model composition approaches, MODEL SWARMS offers tuning-free model adaptation, works in low-data regimes with as few as 200 examples, and does not require assumptions about specific experts in the swarm or how they should be composed. Extensive experiments demonstrate that MODEL SWARMS could flexibly adapt LLM experts to a single task, multi-task domains, reward models, as well as diverse human interests, improving over 12 model composition baselines by up to 21.0% across tasks and contexts. Further analysis reveals that LLM experts discover previously unseen capabilities in initial checkpoints and that MODEL SWARMS enable the weak-to-strong transition of experts through the collaborative search process.

## 1 INTRODUCTION

Advancing beyond efforts to train a single, universal large language model (LLM) (Brown et al., 2020; Gemini Team et al., 2023) that shares parameters across all languages and tasks, recent work has increasingly recognized the importance of modularity through *multi-LLM collaboration*, where diverse models interact and complement each other in various ways (Shen et al., 2024c; Feng et al., 2024a; Chan et al., 2024; Du et al., 2024). For example, mixture-of-experts (MoE) relies on the *routing* of queries to various neural sub-components, leveraging the specialized expertise of one model (Masoudnia & Ebrahimpour, 2014; Roller et al., 2021; Pfeiffer et al., 2022; Jiang et al., 2024). Routing to domain-specific experts demonstrates great potential, while no new model/expert is produced in the MoE process. However, challenging real-world tasks often require flexible composition and adaptation to new domains and/or capabilities that go beyond the scope of an existing expert.

Two lines of work aim to extend multi-LLM collaboration beyond routing to compose and produce new adapted models. 1) *Learn-to-fuse* designs trainable components to "glue" experts together into a merged model, then fine-tunes the model with supervised objectives to produce compositional experts (Jiang et al., 2023b; Wang et al., 2024b; Bansal et al., 2024). These approaches often rely on *large training sets* to tune the learnable parts from scratch and hardly offer the *modularity* of seamlessly adding/removing experts. 2) *Model arithmetic* composes LLM experts by conducting arithmetic operations on model weights and/or token probabilities (Ilharco et al., 2023; Yu et al., 2024; Yadav et al., 2024; Mavromatis et al., 2024; Liu et al., 2024). These approaches often come with strong *assumptions* about the available experts and how the desired adaptation should be decomposed (e.g., *lion indoors = lion outdoors + (dog indoors - dog outdoors)* (Ilharco et al., 2023)). As such, a flexible approach that does not rely on excessive tuning data or strong assumptions about existing models is crucial for adapting diverse LLM experts for wide-ranging purposes.

To this end, we propose MODEL SWARMS, where *multiple LLM experts collaboratively search for new adapted models in the weight space*. Inspired by Particle Swarm Optimization (PSO) (Kennedy & Eberhart, 1995), MODEL SWARMS views each LLM expert as a "particle" and defines LLM adaptation as the collaborative movement of particles governed by a utility function representing an adaptation objective. Specifically, to model the proactive search of LLMs instead of passive merging, each expert particle starts with a *location* (model weights) and a *velocity* (direction in the weight space). The velocity is iteratively impacted by *inertia* (the tendency to keep current velocity), *personal best* (the best-found location of a given particle, best/worst meaning the best/worst-performing

Figure 1: We propose MODEL SWARMS, a collaborative search algorithm to adapt LLM experts via swarm intelligence. Guided by *personal best* $\mathbf{p}_i$, *global best* $\mathbf{g}$, and *global worst* $\mathbf{g}_w$, LLM experts update its velocity $\mathbf{v}$ and location $\mathbf{x}$ to explore the weight space and optimize a utility function $f$. The best-found expert (global best $\mathbf{g}$) in the end is retained as the output.

expert on the utility function), and *global best/worst* (the best/worst-found location among all particles), while LLM particles then take a step towards the updated velocity direction. These velocity factors enable LLM particles to chart an independent search path and explore the personal/global best neighborhoods. Thanks to the flexible search methodology, MODEL SWARMS does not need any supervised fine-tuning data or pre-existing knowledge about the LLM experts or the utility function, adapting LLM experts solely through collaborative search and movement guided by any model-to-scalar utility function.

MODEL SWARMS achieves superior performance across four distinct LLM adaptation objectives:

- *Single task*: Optimizing over as few as 200 instances, MODEL SWARMS outperforms 12 model composition baselines by 13.3% across 9 datasets spanning knowledge, reasoning, and safety.
- *Multi-task domain*: Jointly optimizing multiple tasks in medical, legal, scientific, and cultural domains, MODEL SWARMS often produces Pareto-optimal experts than optimizing a single task.
- *Reward model*: Optimizing reward model scores of general and conflicting preferences, MODEL SWARMS offers steerable experts that outperform baselines by up to 14.6% in controllability.
- *Human interest*: On 16 topics evaluated by humans (e.g., electric vehicles and PhD applications), *Model Swarms* produces experts on par or better than existing models in 85% of cases.

Empirical analyses reveal that the diversity of starting experts is crucial, models display emerging capabilities not seen in initial checkpoints, and surprisingly, the best ending particle often did not start as the best. MODEL SWARMS could be accelerated with dropout-like strategies and seamlessly extended to token probability arithmetic for experts with different model architectures. We envision MODEL SWARMS as a versatile framework to reimagine the potential of diverse open models.

## 2 METHODOLOGY

We propose MODEL SWARMS, a collaborative search algorithm to adapt LLM experts via swarm intelligence. We present an overview of MODEL SWARMS in Figure 1 and Algorithm 1.

MODEL SWARMS assumes the access to *various LLM experts* $\{\mathbf{x}_i\}_{i=1}^n$, which could be full models or LoRA adapters (Hu et al., 2022) fine-tuned on diverse tasks and domains publicly available on model-sharing platforms (Wolf et al., 2019). It also requires a *utility function* $f : \mathbf{x} \rightarrow \mathcal{R}$, mapping each expert onto a scalar value that should be optimized for model adaptation. Utility functions could be dataset performance, reward model scores, or human preferences (Section 3).

Inspired by Particle Swarm Optimization (Kennedy & Eberhart, 1995) and evolutionary algorithms in general (Bäck & Schwefel, 1993), MODEL SWARMS employs several terminologies:

- Each LLM expert, or "particle" in the model swarm, has a *location* represented by model weights;
- Each particle has a *velocity*, a direction in the model weight space that should move towards next;
- *Personal best* $\mathbf{p}_i$: the best-found location of $\mathbf{x}_i$ based on utility function $f$ in its search history;
- *Global best* and *worst* $\mathbf{g}$ and $\mathbf{g}_w$: the best/worst location in all of $\{\mathbf{x}_i\}_{i=1}^n$'s search history.

The location and velocity of particles enable the proactive search of LLM experts instead of passive merging, while the personal/global best checkpoints help keep track of good locations and neighborhoods in the weight space to further explore.

---

**Algorithm 1:** Model Swarms

---

**Input:** LLM experts $\{\mathbf{x}_i\}_{i=1}^n$, utility function $f : \mathbf{x} \to \mathcal{R}$; Hyperparameters: swarm size $N$, step length
      $\lambda$, step length schedule $\phi_\lambda$, inertia $\phi_v$, cognitive coefficient $\phi_p$, social coefficient $\phi_g$, repel
      coefficient $\phi_w$, patience $c$, restart patience $c_r$, max iteration $\mathcal{K}$
  // initialize search
pairwise interpolation to populate initial experts $\{\mathbf{x}_i\}_{i=1}^N = \text{populate}(\{\mathbf{x}_i\}_{i=1}^n)$, $N > n$
initialize global best checkpoint $\mathbf{g} \leftarrow \varnothing$, global worst checkpoint $\mathbf{g}_w \leftarrow \varnothing$
**for** $i = 1$ **to** $N$ **do**
    initialize personal best $\mathbf{p}_i \leftarrow \mathbf{x}_i$, velocity $\mathbf{v}_i \leftarrow \text{random}(\{\mathbf{x}_j\}_{j=1}^N) - \mathbf{x}_i$
    **if** $f(\mathbf{x}_i) > f(\mathbf{g})$, $\mathbf{g} \leftarrow \mathbf{x}_i$; **if** $f(\mathbf{x}_i) < f(\mathbf{g}_w)$, $\mathbf{g}_w \leftarrow \mathbf{x}_i$
**end**
  // search!
**for** $k = 1$ **to** $\mathcal{K}$ **do**
    **if** $\mathbf{g}$ did not change in the last $c$ iterations **then** break
    **for** $i = 1$ **to** $N$ parallel† **do**
        randomness factors $r_v, r_p, r_g, r_w \sim \mathcal{U}(0, 1)$
        update velocity $\mathbf{v}_i \leftarrow \frac{1}{\mathcal{C}}[r_v\phi_v\mathbf{v}_i + r_p\phi_p(\mathbf{p}_i - \mathbf{x}_i) + r_g\phi_g(\mathbf{g} - \mathbf{x}_i) - r_w\phi_w(\mathbf{g}_w - \mathbf{x}_i)]$, where
          normalization term $\mathcal{C} = r_v\phi_v + r_p\phi_p + r_g\phi_g + r_w\phi_w$
        update location $\mathbf{x}_i \leftarrow \mathbf{x}_i + \lambda\mathbf{v}_i$
        **if** $f(\mathbf{x}_i) > f(\mathbf{g})$, $\mathbf{g} \leftarrow \mathbf{x}_i$; **if** $f(\mathbf{x}_i) < f(\mathbf{g}_w)$, $\mathbf{g}_w \leftarrow \mathbf{x}_i$; **if** $f(\mathbf{x}_i) > f(\mathbf{p}_i)$, $\mathbf{p}_i \leftarrow \mathbf{x}_i$
        **if** $f(\mathbf{p}_i)$ didn't change in $c_r$ iterations, $\mathbf{x}_i \leftarrow \mathbf{p}_i$ and $\mathbf{v}_i \leftarrow \mathbf{0}$
    **end**
    step length scheduling $\lambda \leftarrow \lambda \times \phi_\lambda$
**end**
**return** g

---

**Step 0. Initialize** To expand the pool of starting experts/particles $\{\mathbf{x}_i\}_{i=1}^n$, MODEL SWARMS employs pairwise crossover with linear interpolation. Concretely, we randomly select two experts $\mathbf{x}_a$ and $\mathbf{x}_b$ from $\{\mathbf{x}_i\}_{i=1}^n$ and sample $t \sim \mathcal{U}(0, 1)$, a new starting particle is obtained by $\mathbf{x}_{new} = t\mathbf{x}_a + (1 - t)\mathbf{x}_b$. Repeat this process for $N - n$ times to expand $\{\mathbf{x}_i\}_{i=1}^n$ into $\{\mathbf{x}_i\}_{i=1}^N$. Expanding the starting particles allows for more trial-and-error bandwidth in the search process.

For each particle $\mathbf{x}_i$, we initialize its velocity as pointing to a random particle $\mathbf{v}_i = \text{random}(\{\mathbf{x}_j\}_{j=1}^N) - \mathbf{x}_i$.* We initialize its personal best as its current location $\mathbf{p}_i = \mathbf{x}_i$ and determine the global best/worst as $\mathbf{g} = \arg\max_{\mathbf{x}} f(\mathbf{x})$ and $\mathbf{g}_w = \arg\min_{\mathbf{x}} f(\mathbf{x})$, $\mathbf{x} \in \{\mathbf{x}_i\}_{i=1}^n$.

**Step 1. Velocity Update** The movement of LLM experts is mainly governed by *velocity* $\mathbf{v}$, directions in the weight space. We posit that the weight neighborhoods of good model checkpoints might be promising to explore (Eilertsen et al., 2020), thus the velocity of particles $\mathbf{v}_i$ is iteratively drawn by personal best $\mathbf{p}_i$, global best $\mathbf{g}$, and repelled by global worst $\mathbf{g}_w$. Concretely:

$$\mathbf{v}_i \leftarrow \frac{1}{\mathcal{C}}\big[r_v\phi_v\mathbf{v}_i + r_p\phi_p(\mathbf{p}_i - \mathbf{x}_i) + r_g\phi_g(\mathbf{g} - \mathbf{x}_i) - r_w\phi_w(\mathbf{g}_w - \mathbf{x}_i)\big]$$

where $\mathcal{C} = r_v\phi_v + r_p\phi_p + r_g\phi_g + r_w\phi_w$ is a normalization term. To dissect this formula:

- The new velocity is the weighted average of four factors: $\mathbf{v}_i$, the particle keeps some of its current velocity (*i.e.* inertia); $(\mathbf{p}_i - \mathbf{x}_i)$, it is drawn towards its personal best; $(\mathbf{g} - \mathbf{x}_i)$, drawn towards the global best; $-(\mathbf{g}_w - \mathbf{x}_i)$, repelled from the global worst. Inertia enables each expert to chart an independent search path, personal/global best terms encourage experts to explore good weight neighborhoods, while the global worst term repels experts to stay clear of bad model checkpoints.

- Hyperparameters – inertia $\phi_v$, cognitive coefficient $\phi_p$, social coefficient $\phi_g$, repel coefficient $\phi_w$, all $\in [0, 1]$ – are configurable and govern how much the search process is impacted by $\mathbf{p}_i$, $\mathbf{g}$, and $\mathbf{g}_w$. In particular, inertia $\phi_v$ has a unique control over *exploration*, where lower $\phi_v$ means more exploration (less impacted by current velocity and more by other models) and vice versa.

- Walk randomness factors $r_v, r_p, r_g, r_w \sim \mathcal{U}(0, 1)$ ensure that the search is not deterministic, boosting particle exploration and are crucial in the collaborative search process (Table 6).

---

*This is to avoid all particles collapsing into the global best $\mathbf{g}$ like a "black hole" and reduce exploration.
†All particles perform velocity and location update in parallel, we omit the time stamp $k$ for brevity.

**Step 2. Weight Update**   Based on velocity $\mathbf{v}$, the weights/locations of LLM experts are updated by taking a step towards $\mathbf{v}$: $\mathbf{x}_i \leftarrow \mathbf{x}_i + \lambda \mathbf{v}_i$, where $\lambda$ is the step length hyperparameter. The updated particles are then evaluated on the utility function $f$ to update $\mathbf{g}$, $\mathbf{g}_w$, and $\{\mathbf{p}_i\}_{i=1}^N$, if necessary.

Since MODEL SWARMS explicitly encourage randomness and exploration, particles might sometimes fail to find desirable locations and stray away: this exploration is made possible by randomness factors $r_v, r_p, r_g, r_w \sim \mathcal{U}(0,1)$, where the impact of personal/global bests are randomly discounted to favor exploration rather than overly quick convergence. We propose to *restart* undesirable particles and give them another chance: concretely, if for particle $i$ the personal best $\mathbf{p}_i$ didn't change in $c_r$ iterations, where $c_r$ is a hyperparameter, we put the particle back to its personal-best location with $\mathbf{x}_i \leftarrow \mathbf{p}_i$ and $\mathbf{v}_i \leftarrow \mathbf{0}$, essentially granting the particle another chance with a relatively good starting point. In this way, MODEL SWARMS strikes a balance between exploration and robustness.

**Step 3. End of Iteration**   If the global best $\mathbf{g}$ hasn't changed in $c$ iterations (patience hyperparameter) or the maximum iteration of $\mathcal{K}$ is achieved, the search process ends. Otherwise the step length $\lambda$ is reduced by a hyperparameter factor $\phi_\lambda$, $\lambda \leftarrow \lambda \times \phi_\lambda$, and goes back to step 1. In the end, the global best expert $\mathbf{g}$ is returned as the product of MODEL SWARMS.

## 3   EXPERIMENT SETTINGS

**Models and Implementation**   We implement a prototype of MODEL SWARMS with GEMMA-7B (*google/gemma-7b-it*) (Gemma Team et al., 2024) in the main paper, while we also employ other LLMs such as MISTRAL-7B (Jiang et al., 2023a) in Table 8. We create a pool of 10 initial experts/particles by fine-tuning GEMMA-7B separately on the 10 SFT data domains‡ in Tulu-v2 (Ivison et al., 2023) with LoRA (Hu et al., 2022). We fine-tune for 5 epochs with a starting learning rate of 2e-4 and effective batch size of 32 by default. For MODEL SWARMS searches, we employ $N = 20$, $\phi_\lambda = 0.95$, $p = 10$, $p_r = 5$, $\mathcal{K} = 50$, while running grid search over other hyperparameters and report the best-found expert based on utility function $f$.

**Baselines**   We compare with 12 model composition baselines in three categories.

- **Trivial composition**, 1) *Best Single* expert, essentially $\arg\max_{\mathbf{x}} f(\mathbf{x})$ for $\mathbf{x} \in \{\mathbf{x}_i\}_{i=1}^n$; 2) *Data Merge*, where the 10 SFT data domains in Tulu-v2 are merged to train one single expert; 3) *Prediction Merge*, where the predictions of $\{\mathbf{x}_i\}_{i=1}^n$ are ensembled via plurality vote (if applicable).
- **Static composition**, where the composed expert is independent of the adaptation task/utility function $f$. We evaluate *Uniform Soup* (Wortsman et al., 2022a), *Slerp*, *Dare-Ties* (Yu et al., 2024; Yadav et al., 2024), and *Model Stocks* (Jang et al., 2024).
- **Dynamic composition**, where the composed expert changes based on the utility function $f$. We evaluate *Greedy Soup* (Wortsman et al., 2022a), *Pack of LLMs* (Mavromatis et al., 2024), *cBTM* (Gururangan et al., 2023), *EvolMerge* (Akiba et al., 2024), and *LoraHub* (Huang et al., 2023). These approaches are also guided by the utility function $f$ like MODEL SWARMS.

**Data and Evaluation**   We investigate whether MODEL SWARMS could adapt LLM experts via collaborative search on four types of adaptation objectives and the corresponding utility functions.

- **Single task**: we employ 9 datasets spanning knowledge (MMLU (Hendrycks et al., 2021), MMLU-pro (Wang et al., 2024e), Hellaswag (Zellers et al., 2019)), reasoning (GSM8k (Cobbe et al., 2021), Knowledge Crosswords (Ding et al., 2024), NLGraph (Wang et al., 2024a; Zhang et al., 2024b)), and safety (TruthfulQA (Lin et al., 2022), RealToxicityPrompts (Gehman et al., 2020), AbstainQA (Feng et al., 2024a)). We by default randomly sample 200 and 1000 samples as the validation/test sets: the utility function $f$ is defined as performance on the validation set.
- **Multi-task domain**: in addition to optimizing for one task, models should also be adaptable to an application domain comprising of multiple tasks. We employ 4 such domains and 2 tasks in each domain, specifically medical (MedQA (Jin et al., 2021; Li et al., 2024b) and MedMCQA (Pal et al., 2022)), legal (hearsay and citation prediction classification in LegalBench (Guha et al., 2024)), scientific (SciFact (Wadden et al., 2020) and the STEM subset of MMLU-pro (Wang et al., 2024e)), and culture (the country-based and value-based subtasks of Normad (Rao et al., 2024)). The utility function $f$ is defined as the harmonic mean of performance on the two tasks.

---

‡We replace the *GPT-4 Alpaca* subset with Gemini-distilled Alpaca and remove the *hardcoded* subset.

| | MMLU | | MMLU-pro | | Hellaswag | | K-Crossword | | GSM8k | | NLGraph | | TruthfulQA | | RTPrompts | | AbstainQA | |
|---|---|---|---|---|---|---|---|---|---|---|---|---|---|---|---|---|---|---|
| | val | test | val | test | val | test | val | test | val | test | val | test | val | test | val | test | val | test |
| Best Single | .555 | .537 | .357 | .231 | .605 | .601 | .395 | .346 | .220 | .237 | .540 | .535 | .365 | .308 | .913 | .860 | .020 | .065 |
| Data Merge | .435 | .445 | .300 | .176 | .505 | .527 | .380 | .370 | .080 | .143 | .395 | .423 | .160 | .107 | .880 | .848 | -.090 | -.025 |
| Pred. Merge | .525 | .542 | .414 | .173 | .565 | .586 | .295 | .309 | .075 | .074 | .505 | .502 | .325 | .276 | / | / | / | / |
| Uniform Soup | .525 | .530 | .314 | .206 | .545 | .552 | .290 | .295 | .270 | .352 | .500 | .500 | .395 | .350 | .890 | .875 | -.040 | .003 |
| Slerp | .550 | .559 | .386 | .237 | .560 | .614 | .350 | .309 | .205 | .256 | .520 | .530 | .345 | .313 | .915 | .884 | .070 | .128 |
| Dare-Ties | .560 | .567 | .414 | .230 | .600 | .622 | .410 | .372 | .230 | .307 | .560 | .544 | .380 | .337 | .905 | .867 | .110 | .140 |
| Model Stocks | .545 | .543 | .357 | .221 | .540 | .565 | .320 | .310 | .255 | .350 | .505 | .502 | .400 | .339 | .895 | .873 | .010 | .012 |
| Greedy Soup | .575 | .554 | .371 | .219 | .630 | .596 | .395 | .355 | .255 | .330 | .545 | .530 | .410 | .345 | .916 | .860 | .105 | .014 |
| Pack of LLMs | .515 | .568 | .371 | .235 | .630 | .593 | .375 | .352 | .245 | .327 | .540 | .532 | .370 | .295 | .916 | .861 | -.065 | .095 |
| cBTM | .510 | .506 | .286 | .179 | .510 | .525 | .320 | .284 | .160 | .198 | .410 | .398 | .360 | .314 | .885 | .842 | -.060 | -.029 |
| EvolMerge | .545 | .548 | .371 | .229 | .565 | .574 | .300 | .293 | .320 | .354 | .510 | .506 | .395 | .340 | .896 | .870 | .050 | .018 |
| LoraHub | .555 | .554 | .386 | .231 | .570 | .573 | .345 | .291 | .315 | .354 | .565 | .568 | .425 | .359 | .903 | .885 | .100 | .064 |
| Model Swarms | **.605** | **.583** | **.443** | **.254** | **.675** | **.652** | **.470** | **.428** | **.395** | **.459** | **.730** | **.672** | **.455** | **.392** | **.957** | **.956** | **.200** | **.175** |

Table 1: Performance on the validation and test sets of the 9 datasets. Best in **bold** and second-best in underline. MODEL SWARMS outperforms TRIVIAL, STATIC, and DYNAMIC baselines by 13.3% on average and works best on the middle three reasoning tasks with an improvement of 21.0%.

- **Reward model**: we employ three reward models (RMs) to adapt to general and conflicting preferences: a general RM (*internlm/internlm2-7b-reward* (InternLM Team, 2023)) and we train two conflicting RMs, verbose-RM and concise-RM, adapted from the general RM and each preferring longer and more comprehensive *vs.* shorter and straight-to-the-point responses, studying whether MODEL SWARMS and baselines could offer steerability in model behavior and adapt to pluralistic human preferences (Sorensen et al., 2024). We sample 200 instructions from AlpacaFarm (Dubois et al., 2024) as the validation set and 550 instructions from AlpacaFarm and Koala (Geng et al., 2023) as the test set. $f$ is defined as the RM scores on the validation set. We additionally employ PPO (Schulman et al., 2017) and DPO (Rafailov et al., 2024) as baselines to see if MODEL SWARMS offers a better use of RMs with as few as 200 instructions.

- **Human interest**: in addition to preferences represented by reward models, it is crucial to adapt LLM experts directly to *human*: their preferences, personalized needs, and interest domains. Specifically, 13 human annotators nominated 16 interest domains (e.g., electric vehicles and PhD applications), we then employ GEMINI-PRO to synthesize 25:25 instructions in each domain as validation/test set. $f$ is defined as LLM-as-a-judge (Zheng et al., 2023) 1-10 scores with Gemini on the validation set, while we evaluate the adaptation to human interest topics on three fronts: improvement in $f$ scores, improvement in factuality with Facts&Evidence (Boonsanong et al., 2024), and human evaluation win rate comparing pre-swarm and post-swarm responses.

## 4 RESULTS

**Single Task** We present the performance of MODEL SWARMS and baselines on 9 datasets in Table 1. MODEL SWARMS achieves state-of-the-art performance on all 9 tasks. It outperforms the second-strongest baseline by 13.3% on average and up to 29.7% on the GSM8k dataset. The average improvement on reasoning tasks (middle three, 21.0%) is higher than on knowledge (first three, 4.9%) and safety (last three, 14.1%) tasks, indicating MODEL SWARMS' versatility and unique strength in adapting

| | Medical | | Legal | | Science | | Culture | |
|---|---|---|---|---|---|---|---|---|
| | MedQA | MedMC | Hearsay | Cite. | SciFact | STEM | Country | Value |
| Best Single | .423 | .432 | .638 | .500 | .545 | .171 | .544 | .585 |
| Data Merge | .361 | .346 | .596 | .509 | .570 | .148 | .468 | .587 |
| Uniform Soup | .403 | .428 | .521 | .491 | .680 | .146 | .481 | .504 |
| Slerp | .424 | .431 | .610 | .528 | .729 | .167 | .514 | .528 |
| Dare-Ties | .424 | .437 | .631 | .537 | .724 | .171 | .534 | .546 |
| Model Stocks | .409 | .432 | .543 | .444 | .727 | .159 | .507 | .540 |
| Greedy Soup | .427 | .439 | .631 | .472 | .680 | .161 | .526 | .553 |
| Pack of LLMs | .418 | .435 | .521 | .545 | .699 | .165 | .500 | .533 |
| cBTM | .380 | .342 | .463 | .463 | .709 | .165 | .527 | .474 |
| EvolMerge | .415 | .431 | .532 | .491 | .667 | .163 | .503 | .527 |
| LoraHub | .405 | .429 | .588 | .536 | .711 | .159 | .541 | .557 |
| Model Swarms | **.443** | **.457** | **.702** | **.602** | **.743** | **.188** | **.559** | **.603** |

Table 2: Test set performance on the 8 tasks across 4 domains in multi-task domain adaptation. Best in **bold** and second-best in underline. MODEL SWARMS outperforms all 12 baselines by 5.7% on average across datasets.

to diverse reasoning-intensive contexts due to stronger generalization on reasoning problems. DYNAMIC merging baselines achieve 11 out of all 18 second-place positions, with an average performance 15.6% and 2.1% higher than TRIVIAL and STATIC approaches. This indicates that how to compose models is task-dependent, while MODEL SWARMS advances the task-specific adaptation via multi-LLM collaborative search and further outperforms DYNAMIC approaches by 20.8%.

| Interest Topic | LLM Judge | Factuality | Human Eval Win Rate | Interest Topic | LLM Judge | Factuality | Human Eval Win Rate |
|---|---|---|---|---|---|---|---|
| south america | 6.28 → **7.32** | .50 → **.55** | | sandbox games | 5.84 → **6.88** | .48 → **.62** | |
| legal AI | 6.36 → **7.60** | .46 → **.48** | | cartoons | 6.40 → **7.48** | .50 → **.72** | |
| aircraft AI | 6.52 → **7.76** | .47 → **.52** | | music instrument | 6.48 → **7.52** | .73 → **.76** | |
| phd application | 6.16 → **7.52** | .39 → **.45** | | olympics | 5.92 → **6.92** | .77 → **.79** | |
| asian food | 6.28 → **7.20** | .44 → **.47** | | economics | 6.32 → **7.56** | .41 → **.48** | |
| finance | 6.72 → **7.76** | .42 → **.53** | | electric vehicles | 6.56 → **7.64** | .40 → **.42** | |
| luxury cars | 6.40 → **7.60** | .12 → **.30** | | plastic | 6.28 → **7.40** | .44 → **.53** | |
| social network | 6.56 → **7.60** | .43 → **.48** | | us tourism | 6.12 → **7.28** | .51 → **.60** | |

Table 3: LLM-as-a-judge scores with Gemini-Flash, factuality scores with Facts&Evidence (Boon-sanong et al., 2024), and human eval win rates comparing pre- and post-MODEL SWARMS across 16 human interest domains. Colors indicate WIN, TIE, and LOSE. MODEL SWARMS improve both scores by 17.6% and 17.0% on average, while achieving 70.8% average win rate across 16 topics.

**Multi-Task Domain** We present test set performance across 8 tasks and 4 domains in Table 2. Although the multi-task domain adaptation setting is more challenging, MODEL SWARMS still leads to an average improvement of 5.7% over baselines. Specifically, in the legal domain, we see the most substantial performance improvement (11.3% and 10.5%). In addition, we discover that MODEL SWARMS produces *Pareto-Optimal experts* (Figure 10), i.e., jointly optimizing two tasks in one shared domain often outperforms only adapting to one single task.

**Reward Model** We present the reward model scores on validation and test set instructions in Table 4. MODEL SWARMS outperforms all 14 baselines by 6.7% on average, including PPO and DPO, in the low-data adaptation regime with 200 instructions only. Importantly, while on par with alignment methods on general RM, MODEL SWARMS offers impressive steerability to adapt to diverse/conflicting user preferences, instantiated here as verbose vs. concise. While most baselines could only reflect one but not the other (e.g. SLERP is good on *verbose* but bad on *concise*), MODEL SWARMS achieves state-of-the-art performance on both verbose and concise RMs, indicating that the flexible collaborative search methodology presents a viable solution for aligning to diverse and pluralistic human preferences (Wang et al., 2023; Sorensen et al., 2024; Feng et al., 2024b).

| | General RM | | Verbose RM | | Concise RM | |
|---|---|---|---|---|---|---|
| | val | test | val | test | val | test |
| BEST SINGLE | .559 | .562 | .650 | .642 | .533 | .490 |
| DATA MERGE | .457 | .445 | .527 | .550 | .430 | .396 |
| UNIFORM SOUP | .625 | .612 | .575 | .572 | .550 | .540 |
| SLERP | .327 | .316 | .608 | .611 | .220 | .204 |
| DARE-TIES | .415 | .403 | .607 | .604 | .335 | .315 |
| MODEL STOCKS | .615 | .593 | .562 | .567 | .555 | .526 |
| GREEDY SOUP | .621 | .609 | .650 | .649 | .578 | .553 |
| PACK OF LLMs | .614 | .609 | .625 | .641 | .547 | .529 |
| cBTM | .567 | .559 | .601 | .591 | .466 | .463 |
| EVOLMERGE | .619 | .607 | .575 | .570 | .554 | .536 |
| LORAHUB | .621 | .603 | .616 | .619 | .589 | .561 |
| PPO | .628 | .574 | .618 | .641 | .536 | .527 |
| DPO | .627 | .617 | .681 | .682 | .611 | .599 |
| MODEL SWARMS | **.646** | **.621** | **.780** | **.770** | **.651** | **.639** |

Table 4: Reward model scores on the validation and test instruction sets. Best in **bold** and second-best in underline. MODEL SWARMS flexibly adapts to both general and steerable preference RMs, improving by 6.7% on average.

| Topic | Query | Pre-Swarm Response | Post-Swarm Response |
|---|---|---|---|
| legal AI | How can NLP technology enhance legal documents? | Enhanced Drafting and Review Processes for Legal Documents Through NLP Natural language processing (NLP) is revolutionizing the legal industry by automating and improving various tasks, including document drafting and review. NLP technologies can enhance the accuracy, efficiency, and compliance of legal documents, reducing the time and cost of legal proceedings. NLP technology could generate many sentences, documents, and content that could be helpful to legal professionals. (score: 5/10) | NLP technology can significantly enhance legal documents, ensuring precision and compliance. Here's how: 1. Extracting Key Information: NLP can analyze legal documents and extract relevant information like contracts, agreements, and regulations. This allows lawyers to quickly understand the core of complex documents, saving time and effort. 2. Identifying Legal Entities and Relationships: NLP can identify and categorize legal entities and their relationships within documents. This facilitates accurate document organization and analysis. (score: **8**/10) |
| music | What are some stylistic differences between bass in jazz versus rock music? | Sure, here are some distinct stylistic differences between the use of bass in jazz versus rock music: The bass is often used to play the melody, rather than just the root notes. The bass is used to play chords, rather than just root notes. The bass is used to play a variety of rhythm patterns, including syncopation, swing, and waltz. The bass is used to play a variety of dynamic changes, including volume changes, pitch changes, and tempo changes. (score: 5/10) | Stylistic differences are evident in the instrument's role, playing techniques, and overall musical context. Melodical Focus: The bass often takes the lead, playing intricate melodies and improvisational lines. Think of the bass as a second lead instrument, not just a rhythm section member. Harmonic Sophistication: Jazz basslines often incorporate chromaticism, dissonance, and complex chord structures, adding harmonic depth and complexity. Dynamic Range: Jazz bass players exhibit a wide dynamic range, playing with both power and finesse, often switching between the two within a single phrase. (score: **7**/10) |

Table 5: Excerpts of LLM-generated texts in legal AI and music domains with LLM-as-a-judge scores. Post-swarm LLM experts generate more detailed, comprehensive, and helpful responses.

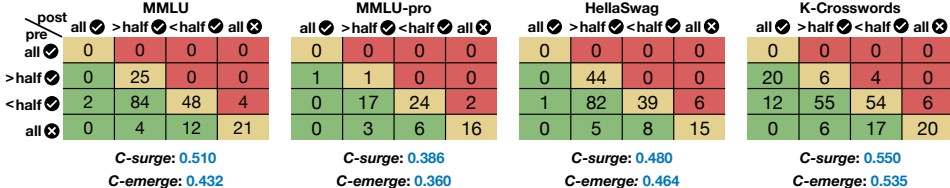

Figure 2: The number of problems in each correctness level for experts before and after MODEL SWARMS across four datasets, along with *C-surge* and *C-emerge* metrics. Cell colors indicate UP, SAME, and DOWN changes in correctness levels. MODEL SWARMS discover new capabilities and skills through collaborative search evident in the 44.8% average *C-emerge*, solving 44.8% of previously "impossible" problems for all initial model checkpoints.

**Human Interest** We present the comparison between pre- and post-MODEL SWARMS experts in the 16 human-nominated interest domains in Table 3 and examples in Table 5. Through adaptation with MODEL SWARMS, experts improve 17.6% in LLM-as-a-judge scores and 17.0% in factuality scores on average when discussing the 16 topics and domains. Most importantly, human evaluation reveals that MODEL SWARMS features a 70.8% win rate against initial experts on average, in particular, with an impressive 96% win rate in the two most successful domains while still maintaining 44%:28%:28% on the unfamiliar and most challenging topics. This indicates that MODEL SWARMS outputs are consistently preferred by both automatic metrics and human users, indicating MODEL SWARMS' great potential to produce domain-specialized and community-specific LLM experts.

## 5 ANALYSIS

**Correctness Emergence** In the collaborative search process, are LLM experts simply transferring existing capabilities from one model to another, or are they discovering new skills and expertise for adaptation? Specifically, there are four *correctness levels* for a question and the pool of LLM experts: ① the answers of experts are *all wrong*; ② *less than half correct*; ③ *more than half correct*; and ④ *all correct*. The correctness level for a question could change between the pre- and post-MODEL SWARMS experts (e.g. (① → ③) indicates that none of the experts answered correctly initially, but after MODEL SWARMS optimization more than half answered correctly.) We define two metrics, *correctness surge* (*C-surge*) and *correctness emergence* (*C-emerge*):

$$C\text{-}surge = \frac{\sum_{j>i} |\,ⓘ \rightarrow ⓙ\,|}{\sum_{i,j \in [1,4]} |\,ⓘ \rightarrow ⓙ\,|}, \quad C\text{-}emerge = \frac{\sum_{j>1} |\,①\rightarrow ⓙ\,|}{\sum_{j \in [1,4]} |\,① \rightarrow ⓙ\,|}$$

where *C-surge* indicates the percentage of questions with an increased correctness level after MODEL SWARMS, and *C-emerge* quantifies that out of all initially type-① questions, how much was correctly answered by at least one expert after MODEL SWARMS. Figure 2 illustrates the changes in correctness levels: MODEL SWARMS achieves an average *C-surge* of 48.2% across the four datasets, indicating broad expert improvements. An interesting observation is that MODEL SWARMS achieves 36.0% to 53.5% *C-emerge*, indicating that the collaborative search surfaced new skills and capabilities in experts that solved 36.0% to 53.5% previously "impossible" problems for all initial experts.

**Diamond in the Rough** We observe that in MODEL SWARMS searches, *the experts that ended as the best didn't necessarily start as the best*. We illustrate this in Figure 3: for particles that ended with the highest utility in a swarm, what was its ranking based on $f$ before the search? Averaged across the four datasets, we found that only 10.4% of the ending-best particles also started as the best (#1), while surprisingly the bottom half of the starting experts were able to rise to the top in 56.9% of the MODEL SWARMS searches. This indicates that weak experts are not inherently less effective but maybe simply not fully adapted to the

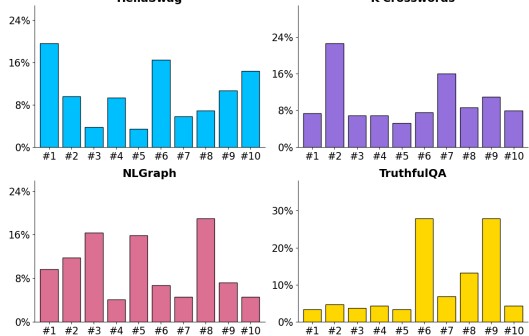

Figure 3: The distribution of starting rankings for experts that ended as the best. 89.6% did *not* start as the best and 56.9% started in the bottom half.

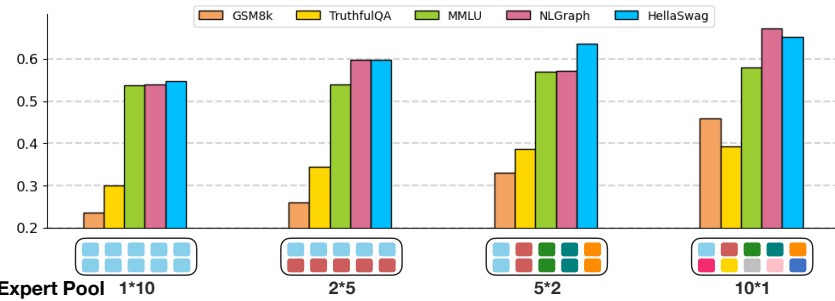

Figure 4: MODEL SWARMS with increasing levels of diversity in initial LLM experts. Results show a general upward trend and a 35.3% increase on average from the least to most diverse initial experts.

task/domain/context of use: they are rightfully *diamond in the rough* and MODEL SWARMS enables the weak-to-strong transition that activates their implicit expertise to produce strong adapted LLM experts. This also indicates that the global best status is switching between experts frequently, suggesting that models are vibrantly and collectively improving and the top spot is constantly overtaken.

**Diversity Matters**   MODEL SWARMS rely on a pool of LLM experts to run the collaborative search algorithm and produce adapted models. Amid the 922,559 models[§] publicly available on Huggingface (Wolf et al., 2019), what models should we select? Specifically, do we need homogeneous model checkpoints or diverse specialized experts? To this end, we conduct a controlled experiment: we take $a$ distinct initial experts (Section 3) and repeat each for $b$ times to result in the starting swarm (denoted as $a \times b$) while controlling $a * b$ as a constant, then employ MODEL SWARMS to adapt them to a task/dataset. We present the results for $1 \times 10$, $2 \times 5$, $5 \times 2$, and $10 \times 1$ in Figure 4, from the least diverse to the most diverse. Experiments demonstrate a consistent upward trend with the increase in expert diversity, while $10 \times 1$ outperforms $1 \times 10$ by 35.3% averaged across the five datasets. This indicates that *diversity matters*, that the success of MODEL SWARMS hinges on the collaborative search of a diverse and wide-ranging pool of initial experts.

**Different Model Architectures with Token Swarms**   The default MODEL SWARMS algorithm operates on *model weights*, *i.e.* the arithmetic operation of updating particle velocity and location is instantiated with model parameter values (*weight swarms*). What if we need to compose experts fine-tuned from *different* base architectures? Instead of model weights, the swarm intelligence arithmetic could be seamlessly carried out on *token probability distributions* for *token swarms*.

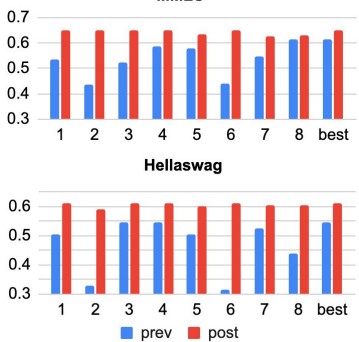

Figure 5: Performance of the token probability variation.

Concretely, the $n$ experts start with a location matrix as an identity matrix $\mathbf{L} = \mathbf{I}_{n \times n} = [\mathbf{l}_1, \cdots, \mathbf{l}_n]$, where the $i$-th row $\mathbf{l}_i$ denotes the location of particle $i$: a one-hot vector of 0s and the $i$-th value is 1. For text generation, denoting the next-token probability distribution of expert $i$ as $\mathbf{t}_i$, expert $i$'s adjusted token probability becomes $\mathbf{t}'_i = \sum_{j=1}^{n} \mathbf{l}_{i,j} \mathbf{t}_j$ and decode text with $\mathbf{t}'_i$. In the beginning, $\mathbf{t}'_i = \mathbf{t}_i$ as the expert focuses solely on its own token probabilities. After running updates of location and velocity in the $n$-dimensional search space (Algorithm 1), $\mathbf{t}'_i$ becomes a composition of $\mathbf{t}$ across experts to optimize $f$. This resembles the collaborative decoding paradigms in existing research (Liu et al., 2024; Shen et al., 2024a), while how to compose the distributions are auto-discovered.

We run a prototype of *token swarms* with 4 experts fine-tuned from GEMMA-7B and 4 from MISTRAL-7B, featuring different model architectures. We present the pre- and post-swarm performance of the 8 experts in Figure 5. All 8 experts become better regardless of model architecture and the global best increased 5.7% and 11.9% on the two datasets. We envision a

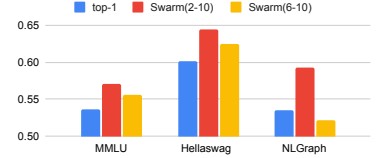

Figure 6: The performance of *swarm*(2-10) and *swarm*(6-10).

---

[§]Accessed on Sept 8, 2024.

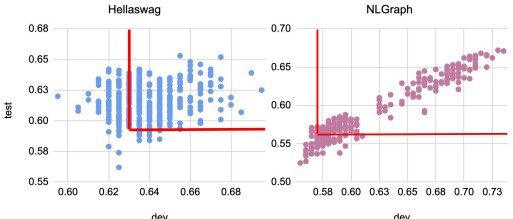

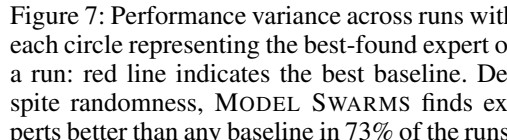

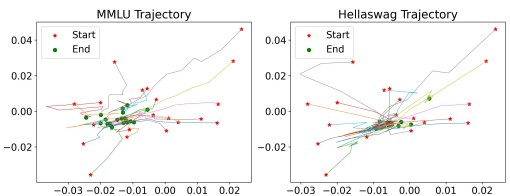

Figure 7: Performance variance across runs with each circle representing the best-found expert of a run: red line indicates the best baseline. Despite randomness, MODEL SWARMS finds experts better than any baseline in 73% of the runs.

Figure 8: Visualization of the model search trajectories on two datasets, where each colored line represents the movement in weight space for one LLM expert. Diverse experts collaboratively search for composition and converge to adapted models in the weight space.

full-blown implementation and analysis as well as some modifications to the *token swarms* variant as important future work.

**Collaboration of Weak > Strong** When we don't have strong starting experts to begin with, would MODEL SWARMS enable the collaboration of weaker models to beat the strong? We investigate this by sorting $\{\mathbf{x}_i\}_{i=1}^n$ by utility $f$, withhold the top-1 model and see whether the collaboration of the remaining experts would surpass it, i.e., whether $\mathrm{Swarm}(\{\mathbf{x}_i\}_{i=2}^n) > \mathbf{x}_1$. We also evaluate the collaboration of the bottom half, $\mathrm{Swarm}(\{\mathbf{x}_i\}_{i=n/2}^n)$, and present performance in Figure 6. It is demonstrated that the collaboration of weak models could beat the top-1 expert, with an average improvement of 35.4% across the four datasets. The collaboration of the bottom half also outperforms the top-1 in 2 out of 3 datasets, suggesting that MODEL SWARMS enables the *weak-to-strong* (Burns et al., 2024) transition of LLM experts through collaborative search.

**Randomness Ablation** We explicitly enable randomness in MODEL SWARMS, with the hope of boosting exploration and adaptation. Specifically, randomness comes in three steps:

1. random interpolation to grow initial experts $\{\mathbf{x}_i\}_{i=1}^N = \mathrm{populate}(\{\mathbf{x}_i\}_{i=1}^n)$, $N > n$;

2. random starting velocity $\mathbf{v}_i \leftarrow \mathrm{random}(\{\mathbf{x}_i\}_{i=1}^N) - \mathbf{x}_i$;

3. random velocity update weights $r_v, r_p, r_g, r_w \sim \mathcal{U}(0,1)$;

We conduct an ablation study where we disable the three randomness in Table 6. We find that the three randomness factors all contribute to model performance across the four datasets, while the deterministic variant (no 1 & 2 & 3) would result in a 23.5% drop on average.

| SETTING | MMLU | Hellaswag | NLGraph | AbstainQA |
|---|---|---|---|---|
| FULL | **0.583** | **0.652** | **0.672** | **0.175** |
| w/o 1 | 0.504 | 0.587 | 0.530 | 0.099 |
| w/o 2 | 0.516 | 0.615 | 0.523 | 0.049 |
| w/o 3 | 0.544 | 0.611 | 0.547 | 0.147 |
| w/o 1 & 2 | 0.561 | 0.601 | 0.611 | 0.091 |
| w/o 1 & 3 | 0.536 | 0.600 | 0.527 | 0.055 |
| w/o 2 & 3 | 0.554 | 0.606 | 0.532 | 0.082 |
| w/o 1 & 2 & 3 | 0.528 | 0.611 | 0.541 | 0.072 |

Table 6: Performance with randomness in 1) initial interpolation, 2) starting velocity, and 3) velocity update removed.

We further visualize performance variance due to these randomness factors. We run for up to 200 times, and present the val/test performance variance in Figure 7. Despite the randomness, MODEL SWARMS is consistently producing adapted experts better the best baseline, outperforming it in 73% of runs.

**Visualizing Search Trajectory** Since the same arithmetic is applied equally to all model parameters, we could visualize the search trajectory of LLM experts by plotting any two parameter values. Figure 8 demonstrates that starting as diverse LLM experts, models collaboratively search in the weight space and converge to a weight area that best optimizes the objective $f$.

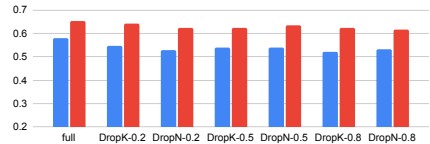

Figure 9: Performance with Drop-K and Drop-N, speeding up MODEL SWARMS by up to 80% with only a 6.0% drop.

**Accelerating with Dropout-K/N** By default, the utility function $f$ is evaluated for every LLM expert at every single iteration. To speed up, we propose Drop-K $d_k$ and Drop-N $d_n$: randomly

skipping model evaluation in $d_k\%$ of iterations or for $d_n\%$ of experts. We evaluate $\{d_k, d_n\} = \{0.2, 0.5, 0.8\}$ and present model performance in Figure 9. With an speed up of up to 80% comes with only a slight performance drop of 6.0% on average, indicating that Drop-K and Drop-N present two helpful strategies to reduce the computational costs of MODEL SWARMS while maintaining good expert utility.

## 6 RELATED WORK

**Composing Diverse LLM Experts**  In addition to developing gargantuan general-purpose LLMs, increasing research focus on the composition of multiple models. *Mixture-of-experts (MoE)* models (Jiang et al., 2024; Lin et al., 2024) and methods (Roller et al., 2021; Lewis et al., 2021; Kudugunta et al., 2021; Pfeiffer et al., 2022; Du et al., 2022; Gururangan et al., 2022; Shen et al., 2024b) are one of the most noted paradigms in composing models, where different approaches vary on parallel neural components (Zhang et al., 2022; Li et al., 2022), routing mechanisms (Zhou et al., 2022; Dai et al., 2022), and expert partition (Gururangan et al., 2023; Jang et al., 2023). More recently, *learn-to-fuse* approaches propose to "glue" experts together with trainable modules (Bansal et al., 2024), adapters (Wang et al., 2024b), or even LLMs (Jiang et al., 2023b): these approaches often need substantial supervised data and might not be modular to seamlessly add/remove experts. In addition, *static model arithmetic* approaches propose to compose experts by performing arithmetic on model weights and token probabilities to reconcile sign differences (Yu et al., 2024; Yadav et al., 2024), simulate tuning effects (Liu et al., 2024), and induce compositional capabilities (Ilharco et al., 2023), and more (Davari & Belilovsky, 2023; Jang et al., 2024; Deep et al., 2024; Zheng et al., 2024). In comparison, *dynamic model arithmetic* proposes to merge models guided by an objective function, employing perplexity heuristics (Mavromatis et al., 2024), evolutionary methods (Akiba et al., 2024), and more (Wortsman et al., 2022a; Huang et al., 2023; Gururangan et al., 2023). Most of these model arithmetic approaches often rely on strong *assumptions* about the experts how they should be composed (e.g. *lion indoors = lion outdoors + (dog indoors - dog outdoors)* (Ilharco et al., 2023)). In contrast, MODEL SWARMS presents a modular, assumption-free, and flexible approach to compose and adapt diverse LLM experts guided by as few as 200 data instances.

**Evolutionary Algorithms and LLMs**  MODEL SWARMS is in part inspired by particle swarm optimization (PSO) (Kennedy & Eberhart, 1995), an evolutionary algorithm (EA) solving optimization problems. This echoes a recent and contemporary uptake of EAs, especially genetic algorithms (GAs) in ML/LLMs (Zhao et al., 2023; Lange et al., 2023; Wu et al., 2024; Chao et al., 2024; Lange et al., 2024). EvolMerge (Akiba et al., 2024) seeks to compose a math LLM and a Japanese LLM through discovering better weight/layer and data flows guided by genetic algorithms. PromptBreeder (Fernando et al., 2024) seeks to search for specialized LLM prompts by maintaining a prompt population and conducting LLM-based crossover and mutation to produce better prompts, resembling GA processes. EvoPrompt (Guo et al., 2024a) also follows similar concepts of applying GAs for prompt optimization. We see two key differences between MODEL SWARMS and this line of existing research: most methods focus on improvements in prompt/data engineering (Fernando et al., 2024; Guo et al., 2024a), while MODEL SWARMS seek to adapt LLMs by changing model weights and inducing new expert capabilities (Figure 2), which is more fundamental and offers greater headroom for improvement; existing EA applications mostly employed genetic algorithms that necessitate much hand-crafted rules (Lambora et al., 2019) (how should two prompts/models crossover to produce new ones, how to mutate, etc.), while MODEL SWARMS is inspired by swarm intelligence that come with little to no manual engineering in the composition and collaboration of models.

## 7 CONCLUSION

We propose MODEL SWARMS, a collaborative search algorithm to flexibly adapt diverse LLM experts to wide-ranging purposes. Guided by personal and global best-found locations, LLM experts explore to optimize utility functions representing various adaptation objectives. Extensive experiments demonstrate that MODEL SWARMS outperforms three categories of 12 model composition baselines by up to 21.0% across four types of model adaptation. Further analysis reveals that MODEL SWARMS help discover new skills in the collaborative search process and bring out the best and implicit expertise of weak models for weak-to-strong expert transition.

## LIMITATIONS AND ETHICS STATEMENT

MODEL SWARMS assumes access to a pool of initial experts for collaborative search to adapt language models. On one hand, it might be challenging to select the right pool of LLMs while we present evidence that the diversity of initial experts is crucial to MODEL SWARMS' successes (Figure 4); On another hand, MODEL SWARMS require the update of all experts at each iteration, which might be computationally challenging. We provide time/space complexity analysis in Appendix B and present a preliminary dropout-like acceleration scheme in Figure 9. MODEL SWARMS is uniquely suited to low-data contexts where only a few hundred examples are readily available to serve as the utility function $f$.

MODEL SWARMS aims to *adapt* language models based on their existing expertise rather than *memorizing* new information that was never seen in the training of these experts. While theoretically by changing model weights experts could pick up new information, our preliminary experiments with perplexity as the utility function, a proxy for memorization, indicates that MODEL SWARMS could not reliably optimize perplexity. We envision that temporal updates could be enabled by employing retrieval augmentation (Chen et al., 2023b; Jiang et al., 2023c; Shi et al., 2024; Wang et al., 2024f) over unseen documents in conjunction with MODEL SWARMS.

MODEL SWARMS by default operate on the model weight space, enabling the collaborative search and movement of LLM experts in terms of model parameters. While this paradigm is incompatible with a pool of experts with heterogeneous model architectures, we propose *token swarms* and demonstrate its preliminary success in Figure 5. We highlight the trade-off between *weight swarms* and *token swarms*: weight swarms induces more fundamental change of model capabilities through weight changes, but it would require all experts to share the same architecture; *token swarms* is much more flexible in expert architectures, but only changes the composition of token probabilities without touching on the model's parametric capabilities. We expect a full implementation and adaptations to the *token swarms* variant as important future work.

Unsuccessful MODEL SWARMS searches might be confined to a local minimum without broad exploration of the desirable weight space. While 1) we take several measures in Algorithm 1 to mitigate this (random starting velocity, walk randomness factors, *etc.*), 2) we observe strong empirical performance of MODEL SWARMS and consistent improvement to the global best **g**, and 3) we visualize the movement of particles in Figure 8 demonstrating its convergence quality, one way to mitigate this concern is by annealing/adding noise to go beyond the local search: changing $r_v, r_p, r_g, r_w \sim \mathcal{U}(0,1)$ to $r_v, r_p, r_g, r_w \sim \mathcal{U}(-0.2,1)$ so that models have a small chance of moving towards the reverse direction and potentially jump out of local minimums.

We would like to highlight the dual-use risk of MODEL SWARMS: thanks to its flexible adaptation strategy by using a model-to-scalar utility function $f$, it also leads to malicious use cases by having malicious $f$s. Some examples could include optimizing the reverse reward model scores, optimizing for lower scores on RealToxicityPrompts (Gehman et al., 2020), optimizing for certain social and political biases (Feng et al., 2023), and more. We argue for the responsible use of the MODEL SWARMS methodology as well as the responsible release of adapted experts.

## REPRODUCIBILITY STATEMENT

We provide all details in the implementation and evaluation of MODEL SWARMS in Appendix C. Specifically, Appendix C contains dataset details and statistics (Table 12), implementation details of MODEL SWARMS, hyperparameter settings, details of all 12 baselines in Section 3, details of all 4 evaluation settings in Section 4, specific prompt texts in Table 13 employed in the human interest objective, and specific human evaluation instructions in Table 14. Upon the final version, we will include a link to a publicly accessible repository with all MODEL SWARMS implementation code, prepossessed data files and resources, adapted model checkpoints, as well as instructions on reproducing our results and using MODEL SWARMS beyond tasks included in this paper.

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

## A DISCUSSION

**Three Key Strengths of MODEL SWARMS** 1) *training-free*: by training-free we mean that the composition of models in MODEL SWARMS doesn't require specific training objectives, loss function, gradient descent, or back propagation. This alleviates data dependency: by using as few as 200 examples MODEL SWARMS could produce better adapted experts, while that is only a bit over 3 batches for training-based approaches with a typical effective batch size of 64. 2) *automatic discovery* or *assumption-free*: instead of dictating the composition of models in *A=B+C-D* formulas, MODEL SWARMS automatically discover better adapted experts through swarm intelligence without making assumptions about experts and how they should be composed. 3) *any adaptation objective*: the collaborative search is only guided by a particle-to-scalar utility function $f$ which could be anything: dataset performance, reward model scores, human interests, and more.

**MODEL SWARMS and Optimization Research** MODEL SWARMS is in part inspired by particle swarm optimization, one algorithm in the very rich literature of optimization research. We don't claim that PSO is the only and best applicable algorithm in the modern LLM world: on the contrary, we invite follow-up works that critically examine how classic optimization techniques, especially for non-convex problems without strong guarantees, could be revived in today's context.

**Non-Neural Reward Models** In Figure 4 we demonstrate that MODEL SWARMS could adapt to preferences represented by neural reward models. However, any model-to-scalar utility function $f$ could work and non-neural RMs are definitely possible: optimizing engagement in social media posts, optimizing click-through rates in online ads, optimizing charity donations when advertising a righteous case. We see many positive (and also negative) possibilities when employing MODEL SWARMS in conjunction with non-neural RMs in social-economic contexts.

**Long vs. Short** In Figure 4 we demonstrate that MODEL SWARMS could steerably adapt to either *verbose RM* or *concise RM*, offering use agency and controllability in model behavior. We discuss the distinctions with two other potential solutions: 1) setting *max_new_tokens*, which might result in cutoffs in generated texts; 2) penalizing *[EOS]* tokens, which might tamper with token probabilities and harm generation quality. For a more on-the-fly steerability, we suggest to separately conduct MODEL SWARMS for two conflicting objectives, then employ an interpolation of the two models with a user-controlled scaler from 0 to 1.

**Resilience to Malicious Experts** There is discussion in multi-agent research about the impact of malicious agents (Huang et al., 2024). However, MODEL SWARMS is robust to malicious experts since the only time a model has influence on others is when it becomes the global best **g**, while an intentionally "bad" model has no chance of becoming **g** on the "good" utility function $f$.

**MODEL SWARMS and Multi-Agent Systems** The role of all "experts" in MODEL SWARMS is *homogeneous*, i.e. they pursue the same goal/adapt to the same objective as represented by utility function $f$. In multi-agent systems (Rame et al., 2022; Zaman et al., 2023; Ainsworth et al., 2023; Chan et al., 2024; Talebirad & Nadiri, 2023; Chen et al., 2023a; Zhang et al., 2024a; Abdelnabi et al., 2024; Kannan et al., 2023; Zeng et al., 2024; Guo et al., 2024b; Sun et al., 2024; Han et al., 2024; Ishibashi & Nishimura, 2024; Wang et al., 2024d; Zhao et al., 2024; Chen et al., 2024c; Hong et al., 2024; Smit et al., 2024; Chen et al., 2024a;b), the agents often have different roles to jointly complete a task, albeit those roles are more or less hand-crafted and especially through prompting. We envision future work on adapting MODEL SWARMS and automatically discovering heterogeneous and collaborative agents that jointly serve a purpose.

**MODEL SWARMS and Model Merging** MODEL SWARMS is both *searching* and *merging* (Wortsman et al., 2022b; Davari & Belilovsky, 2023; Deep et al., 2024; Yang et al., 2024; Wan et al., 2024; Rame et al., 2024; Fu et al., 2024; Ramé et al., 2024; Li et al., 2024a; Tang et al., 2024; Wang et al., 2024c; Du & Kaelbling, 2024): searching in the sense that models are proactively moving in the search space for better experts instead of passively being squashed together, merging in the sense that each resulting model is implicitly an expert taking input from other models and changing its weights accordingly. Contrary to the often "many-to-one" paradigm in model merging research where there is only one merged model, MODEL SWARMS is a "many-to-many" operation that yields

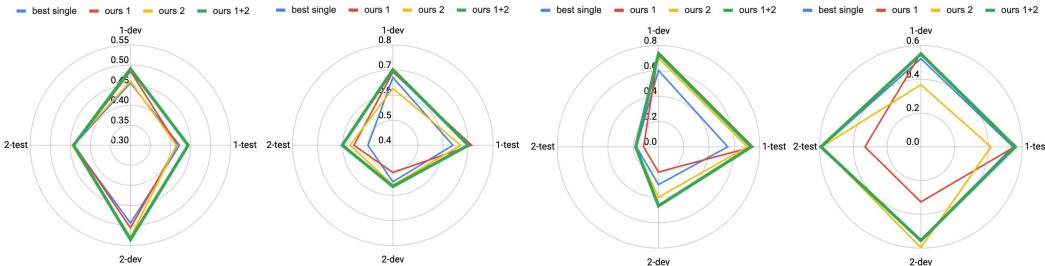

Figure 10: Performance of best single expert, ours only optimizing task 1 or 2, and jointly optimizing tasks 1 and 2. The domains of medical, legal, science, and culture are presented from left to right. MODEL SWARMS produces Pareto-Optimal expert than uni-task optimization.

multiple adapted experts, which open the door for further model merging, another search based on the result of a previous search, MoE routing, and more.

# B   ANALYSIS (CONT.)

**Compositional Capability through Joint Utility Functions**   We investigate whether MODEL SWARMS could adapt to compositional tasks by jointly optimizing two different datasets. Specifically, we investigate *"QA+graph reasoning = multi-hop QA"* with MMLU, NLGraph, and Knowledge Crosswords. We compare the joint utility function of harmonic mean performances against the best single expert without search or searching to optimize one task only. Figure 11 demonstrates that MODEL SWARMS could indeed adapt to compositional tasks by utilizing a combined utility function.

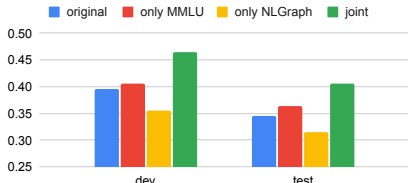

Figure 11: Performance on Knowledge Crosswords with optimizing one dataset or joint task optimization.

**Pareto-Optimal**   In adaptation objective 2: multi-task domains, we argue that the joint optimization of multiple tasks in a single domain might be better than separately optimizing just one. We investigate this by comparing the joint optimization against only optimizing only dataset 1 or 2 in Figure 10. MODEL SWARMS produce mostly Pareto-Optimal experts that's better than optimizing one dataset in most cases.

**Qualitative Examples**   We present qualitative examples for objective 4: human interests, essentially (instruction, pre response, post response) tuples, where human evaluators judge MODEL SWARMS as winning, tying, or losing to initial experts in Tables 15, 16, and 17.

**Ablation Study**   MODEL SWARMS features five major differences from the classic swarm intelligence for LLM optimization: 1) crossover through interpolation and expanding initial expert pool; 2) randomize initial velocity; 3) adding a repel term; 4) adding step length schedule; 5) restarting failing particles. We conduct an ablation study for these five factors in Table 7. It is demonstrated that they are all helpful for model performance, while 1) crossover is most useful.

| SETTING | MMLU | Hellaswag | NLGraph | AbstainQA |
|---|---|---|---|---|
| FULL | 0.583 | 0.652 | 0.672 | 0.175 |
| CROSSOVER, ONLY 15 | 0.527 | 0.604 | 0.534 | 0.093 |
| NO CROSSOVER | 0.504 | 0.587 | 0.53 | 0.099 |
| VELOCITY:BEST | 0.518 | 0.613 | 0.542 | 0.031 |
| VELOCITY:ZERO | 0.516 | 0.615 | 0.523 | 0.049 |
| NO REPEL | 0.534 | 0.631 | 0.534 | 0.025 |
| NO SCHEDULE | 0.517 | 0.611 | 0.536 | 0.095 |
| NO RESTART | 0.532 | 0.628 | 0.532 | 0.131 |

Table 7: Ablation study removing the five modifications to PSO.

**Other LLMs**   To show the generality of MODEL SWARMS, we replace GEMMA-7B with MISTRAL-7B (*mistralai/Mistral-7B-Instruct-v0.3*) and re-run evaluation of adapting to one dataset. Results in Table 8 demonstrates that MODEL SWARMS is general and works regardless of base model.

| | MMLU | | MMLU_pro | | Hellaswag | | K-Crosswords | | GSM8k | | NLGraph | | TruthfulQA | | RTP | | AbstainQA | |
|---|---|---|---|---|---|---|---|---|---|---|---|---|---|---|---|---|---|---|
| | dev | test | dev | test | dev | test | dev | test | dev | test | dev | test | dev | test | dev | test | dev | test |
| best single | 0.385 | 0.433 | 0.257 | 0.146 | 0.545 | 0.550 | 0.415 | 0.364 | 0.190 | 0.303 | 0.335 | 0.325 | 0.380 | 0.353 | 0.898 | 0.873 | -0.130 | 0.081 |
| ours | **0.510** | **0.510** | **0.271** | **0.160** | **0.640** | **0.664** | **0.470** | **0.497** | **0.290** | **0.328** | **0.380** | **0.358** | **0.440** | **0.405** | **0.906** | **0.881** | **0.095** | **0.199** |

Table 8: Performance of single-dataset adaptation with MISTRAL-7B.

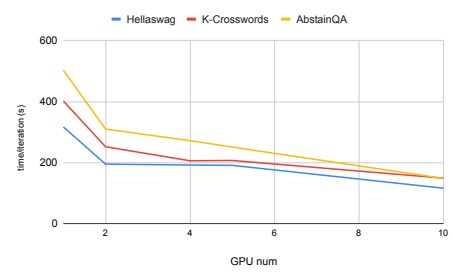
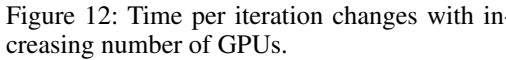

Figure 12: Time per iteration changes with increasing number of GPUs.

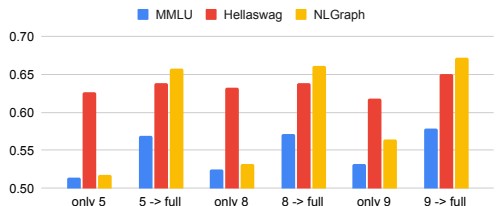

Figure 13: Performance when new experts are injected, from 5 to 10, from 8 to 10, and from 9 to 10, across three datasets. MODEL SWARMS presents the possibility of injecting experts after a search.

**Hyperparameter** We by default run a grid search over several hyperparameters: step length $\lambda$, inertia $\phi_v$, cognitive coefficient $\phi_p$, social coefficient $\phi_g$, and repel coefficient $\phi_w$. We dissect performance with each hyperparameter value in Table 9. It is demonstrated that the changes are minor, thus MODEL SWARMS is largely robust to different hyperparameter configurations.

**Time and Space Complexity** For MODEL SWARMS with $n$ particles, $k$ iterations, the time of validation set inference as $D_1$, the time of test set inference as $D_2$, the time of weight arithmetic as $w$, the probability of global best updating as $a$, then the time complexity is $n(D_1 + 2D_2 + k[(5+a)w + D_1])$ and is $\mathcal{O}(n)$ and $\mathcal{O}(k)$, indicating linear growth with the amount of particles and iterations. For space, the peak storage requirement is $3n + 1$ copies of the LoRA adapters: given the tiny size of adapters, any $n < 100$ should be reasonable.

The implementation of MODEL SWARMS employs multiprocessing, essentially distributing the evaluation of particles to $m$ GPUs with $m$ concurrent threads. We empirically analyze the time complexity of employing 1 to 10 GPUs on our cluster of 16 A100 GPUs with 96 CPU cores with 10 default initial experts. Figure 12 demonstrates that the benefit of more GPUs gradually diminishes due to multiprocessing costs, with 5 GPUs as being optimal on our machine. We show the general trade-off between GPU computation time and multiprocessing time while the exact time per iteration is not meaningful.

| | | MMLU | | NLGraph | | TruthfulQA | |
|---|---|---|---|---|---|---|---|
| | | avg | std | avg | std | avg | std |
| all | | 0.557 | 0.011 | 0.585 | 0.036 | 0.365 | 0.014 |
| inertia | 0.10 | 0.556 | 0.012 | 0.582 | 0.033 | 0.363 | 0.015 |
| | 0.20 | 0.557 | 0.010 | 0.586 | 0.037 | 0.365 | 0.013 |
| | 0.30 | 0.556 | 0.010 | 0.590 | 0.039 | 0.366 | 0.013 |
| cognitive coeff. | 0.10 | 0.557 | 0.010 | 0.584 | 0.041 | 0.362 | 0.015 |
| | 0.20 | 0.558 | 0.011 | 0.588 | 0.037 | 0.364 | 0.013 |
| | 0.30 | 0.556 | 0.009 | 0.590 | 0.041 | 0.367 | 0.014 |
| | 0.40 | 0.556 | 0.011 | 0.587 | 0.033 | 0.365 | 0.014 |
| | 0.50 | 0.557 | 0.012 | 0.578 | 0.028 | 0.365 | 0.014 |
| social coeff. | 0.20 | 0.558 | 0.012 | 0.600 | 0.040 | 0.365 | 0.012 |
| | 0.30 | 0.558 | 0.011 | 0.593 | 0.037 | 0.365 | 0.012 |
| | 0.40 | 0.556 | 0.010 | 0.587 | 0.039 | 0.365 | 0.014 |
| | 0.50 | 0.556 | 0.010 | 0.570 | 0.023 | 0.365 | 0.014 |
| | 0.60 | 0.554 | 0.010 | 0.576 | 0.032 | 0.363 | 0.015 |
| repel coeff. | 0.01 | 0.553 | 0.009 | 0.565 | 0.013 | 0.367 | 0.015 |
| | 0.05 | 0.558 | 0.010 | 0.587 | 0.037 | 0.364 | 0.014 |
| | 0.10 | 0.559 | 0.012 | 0.606 | 0.040 | 0.363 | 0.012 |
| step length | 0.50 | 0.558 | 0.010 | 0.583 | 0.028 | 0.366 | 0.011 |
| | 0.60 | 0.558 | 0.009 | 0.587 | 0.035 | 0.368 | 0.014 |
| | 0.70 | 0.557 | 0.010 | 0.584 | 0.036 | 0.364 | 0.014 |
| | 0.80 | 0.556 | 0.012 | 0.593 | 0.043 | 0.367 | 0.014 |
| | 0.90 | 0.556 | 0.011 | 0.589 | 0.040 | 0.363 | 0.013 |
| | 1.00 | 0.555 | 0.012 | 0.578 | 0.034 | 0.361 | 0.015 |

Table 9: Average model performance under various hyperparameter values.

**Modularity: Adding and Removing Experts** MODEL SWARMS presents a modular multi-LLM collaboration system, where experts could be added/removed from the composition even after a search. For adding experts, since *the only time a particle has an influence on others is when it becomes global best*, we only start the search with the new particle when and if it were to become

|  | MMLU | MMLU_pro | Hellaswag | GSM8k | NLGraph | TruthfulQA |
|---|---|---|---|---|---|---|
| best single | 0.537 | 0.231 | 0.601 | 0.237 | 0.535 | 0.308 |
| SFT | 0.450 | 0.167 | 0.513 | 0.279 | 0.585 | 0.359 |
| Model Swarms | **0.583** | **0.254** | **0.652** | **0.459** | **0.672** | **0.392** |

Table 10: MODEL SWARMS outperforms directly training LLMs on the 200-instance validation set.

**g**. We empirically test this by withholding several experts and injecting others in $5 \rightarrow 10$, $8 \rightarrow 10$, and $9 \rightarrow 10$ settings in Figure 13. Adding experts in this way is generally helpful, while injecting fewer experts is more effective.

As for removing experts, MODEL SWARMS presents a technical guarantee for completely removing an expert and all its influence on other models. We first expand the velocity update term on step $t$:

$$\mathbf{v}_t = r_v \phi_v \mathbf{v}_{t-1} + r_p \phi_p (\mathbf{p}_{t-1} - \mathbf{x}_{t-1}) + r_g \phi_g (\mathbf{g}_{t-1} - \mathbf{x}_{t-1}) - r_w \phi_w (\mathbf{g}_{w,t-1} - \mathbf{x}_{t-1})$$
$$= r_v \phi_v \mathbf{v}_{t-1} + r_p \phi_p \mathbf{p}_{t-1} - (r_p \phi_p + r_g \phi_g - r_w \phi_w) \mathbf{x}_{t-1} + r_g \phi_g \mathbf{g}_{t-1} - r_w \phi_w \mathbf{g}_{w,t-1}$$

The updated location at step $t$ is then:

$$\mathbf{x}_t = \mathbf{x}_{t-1} + \lambda \mathbf{v}_t$$
$$= \mathbf{x}_{t-1} + \lambda \Big[ r_v \phi_v \mathbf{v}_{t-1} + r_p \phi_p \mathbf{p}_{t-1} - (r_p \phi_p + r_g \phi_g - r_w \phi_w) \mathbf{x}_{t-1} + r_g \phi_g \mathbf{g}_{t-1} - r_w \phi_w \mathbf{g}_{w,t-1} \Big]$$
$$= \lambda r_v \phi_v \underline{\mathbf{v}_{t-1}} + \lambda r_p \phi_p \underline{\mathbf{p}_{t-1}} + \Big[ 1 - \lambda (r_p \phi_p + r_g \phi_g - r_w \phi_w) \Big] \underline{\mathbf{x}_{t-1}} + \lambda r_g \phi_g \underline{\mathbf{g}_{t-1}} - \lambda r_w \phi_w \underline{\mathbf{g}_{w,t-1}}$$

Note that $\underline{\mathbf{v}_{t-1}}$, $\underline{\mathbf{p}_{t-1}}$, and $\underline{\mathbf{x}_{t-1}}$ are the property of the particle itself, while $\underline{\mathbf{g}_{t-1}}$ and $\underline{\mathbf{g}_{w,t-1}}$ are the property of potentially other particles. As a result, simply remove the $\mathbf{g}_{t-1}$ and $\mathbf{g}_{w,t-1}$ terms if $\mathbf{g}_{t-1}$ and/or $\mathbf{g}_{w,t-1}$ come from the expert to the removed and normalize the remaining weight terms. For example, if $\mathbf{g}_{t-1}$ andr $\mathbf{g}_{w,t-1}$ are both from the particle to be removed, then:

$$\tilde{\mathbf{x}}_t = \mathcal{C} \Big[ \mathbf{x}_t - \lambda r_g \phi_g \underline{\mathbf{g}_{t-1}} + \lambda r_w \phi_w \underline{\mathbf{g}_{w,t-1}} \Big]$$

where $\mathcal{C} = \dfrac{\lambda r_v \phi_v + \lambda r_p \phi_p + \Big[ 1 - \lambda (r_p \phi_p + r_g \phi_g - r_w \phi_w) \Big] + \lambda r_g \phi_g + \lambda r_w \phi_w}{\lambda r_v \phi_v + \lambda r_p \phi_p + \Big[ 1 - \lambda (r_p \phi_p + r_g \phi_g - r_w \phi_w) \Big]}$ is the weight normalization factor. Starting from $t = 1$ up to $t = \mathcal{K}$ for every $\mathbf{x}$, this removes the specified expert(s) from the composition of other models.

**Search Dynamics** What exactly is happening during a MODEL SWARMS search and how did expert utility change in the process? We visualize the change of each particle as well as the global best in term of utility function $f$ in Figure 15. Experts explore the weight space, their utility scores wax and wane, leading to consistent bumps in global best scores and consequently better adapted language models.

**Prompt Variation** We hypothesize that by optimizing the weights, MODEL SWARMS might offer stronger robustness to minor prompt changes. We employ GEMINI-PRO to *"Please paraphrase the question into 10 versions with minor differences."*, evaluate models on the 10 versions, and calculate the entropy of response distributions as indicators of sensitivity. Figure 14 demonstrates that MODEL SWARMS drastically reduce the sensitivity to minor prompt changes, while still being a bit shy of Gemini-flash/pro levels.

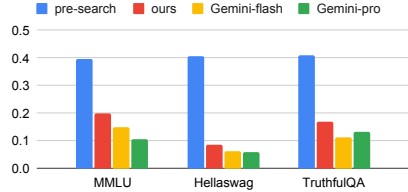

Figure 14: Entropy of model responses indicating sensitivity to 10 prompt versions with minor differences, the lower the better.

**Comparison with Training**   Instead of running MODEL SWARMS, what if we directly fine-tune models on the validation set with its 200 data points? We compare the performance of pre-search best initial expert, post-search global best, and SFT in Table 10. MODEL SWARMS outperforms SFT, indicating that we offer a stronger solution for model adaptation in low-data regimes with as few as 200 instances while SFT might be over-fitting.

**Models of Different Sizes**   We by default employed Gemma-7B in the main experiments: we additionally evaluate Gemma-2B experts on the NLGraph dataset to see if there's an effect with model size in Table 11. The improvement is consistently 20%-30% across 2B and 7B.

## C   EXPERIMENT DETAILS

**Dataset Details**   We employ 20 datasets in total to evaluate MODEL SWARMS and baselines: 9 for objective 1: single task, 8 for objective 2: multi-task domains, 2 for objective 3: reward models, and we synthesize a 16-domain instruction dataset from Gemini (*gemini-1.5-pro-001*) for objective 4: human evaluation. We randomly sample subsets from each dataset and present the statistics in Table 12. We also employ the z-test with the one-tailed hypothesis and present statistical significance test results on the applicable *objective 1: single task* datasets.

| Setting | dev | test |
|---|---|---|
| pre-swarm 2B | 0.315 | 0.330 |
| post-swarm 2B | 0.425 (+34.9%) | 0.420 (+27.2%) |
| pre-swarm 7B | 0.540 | 0.535 |
| post-swarm 7B | 0.730 (+37.0%) | 0.672 (+25.6%) |

Table 11: Experiments on the NLGraph dataset with Gemma-2B and 7B.

**Implementation Details**   For a prototype of MODEL SWARMS, we employ GEMMA-7B (*google/gemma-7b-it*) as the base model checkpoint, then fine-tune it on 10 different supervised fine-tuning domains to obtain 10 initial experts. we specifically employ Tulu-v2 (Ivison et al., 2023), an open collection of instruction-tuning data. We specifically employ the following subsets: flan (Chung et al., 2024), CoT, Open Assistant 1 (Köpf et al., 2024), ShareGPT[†], Code Alpaca (Chaudhary, 2023), LIMA (Zhou et al., 2024), WizardLM Evol-Instruct V2 (Xu et al., 2023), Open-Orca (Lian et al., 2023), and Science Literature (Ivison et al., 2023). We replace the GPT4 Alpaca subset with Gemini Alpaca, distilling generations from *gemini-1.5-pro-001* and remove the hard-coded subset. We employ LoRA fine-tuning (Hu et al., 2022) with a learning rate of 2e-4, cosine learning rate scheduling, effective batch size of 32, warm-up ratio of 0.1, and 5 default training epochs, while we only train for 1 epoch on the large ShareGPT subset. We similarly fine-tune MISTRAL-7B for the experiments in Table 8. We employ greedy decoding for text generation and a maximum new token of 10, 50, 100, or 512 depending on the task.

| Dataset | Source | Size | |
|---|---|---|---|
| | | dev | test |
| MMLU | (Hendrycks et al., 2021) | 200 | 1000 |
| MMLU-pro | (Wang et al., 2024e) | 70 | 1000 |
| K-Crosswords*** | (Ding et al., 2024) | 200 | 1000 |
| Hellaswag* | (Zellers et al., 2019) | 200 | 1000 |
| NLGraph*** | (Wang et al., 2024a) | 200 | 1000 |
| GSM8k*** | (Cobbe et al., 2021) | 200 | 1000 |
| TruthfulQA* | (Lin et al., 2022) | 200 | 617 |
| RealToxicityPrompts*** | (Gehman et al., 2020) | 200 | 1000 |
| AbstainQA** | (Feng et al., 2024a) | 200 | 1000 |
| MedQA | (Li et al., 2024b) | 200 | 1000 |
| MedMCQA | (Pal et al., 2022) | 200 | 1000 |
| Hearsay | (Guha et al., 2024) | 94 | 94 |
| Citation Prediction | (Guha et al., 2024) | 108 | 108 |
| SciFact | (Wadden et al., 2020) | 200 | 532 |
| STEM | (Wang et al., 2024e) | 30 | 473 |
| Normad w/country | (Rao et al., 2024) | 500 | 2000 |
| Normad w/value | (Rao et al., 2024) | 500 | 2000 |
| AlpacaFarm | (Dubois et al., 2024) | 200 | 400 |
| Koala | (Geng et al., 2023) | / | 150 |
| Humen eval | Gemini-synthesized | 16*25 | 16*25 |

Table 12: Statistics of employed datasets. *, **, and *** indicates the improvement on this dataset is statistically significant with $p < 0.1$, $p < 0.05$, and $p < 0.01$ with one-tailed z-test.

**Hyperparameter Settings**   For MODEL SWARMS searches, we employ $N = 20$, $\phi_\lambda = 0.95$, $p = 10$, $p_r = 5$, $\mathcal{K} = 50$, while running grid search over other hyperparameters and report the best-found expert based on utility function $f$. Specifically, we search for $\phi_v \in \{0.1, 0.2, 0.3\}$, $\phi_p \in \{0.1, 0.2, 0.3, 0.4, 0.5\}$, $\phi_g \in \{0.2, 0.3, 0.4, 0.5, 0.6\}$, $\phi_w \in \{0.01, 0.05, 0.1\}$, $\lambda \in$

---

[†]https://sharegpt.com/

$\{0.5, 0.6, 0.7, 0.8, 0.9, 1.0\}$. We run up to 200 to 1000 runs by randomly choosing over these hyperparameter search settings and report the best-found expert on utility function $f$. Though there is randomness, Figure 7 demonstrates that MODEL SWARMS is robust to hyperparameter settings and consistently find experts better than any of the 12 baselines.

**Baseline Details** We employ 12 baselines in total:

- Best single expert: among the 10 initial experts, the expert that performed best on utility function $f$ is evaluated and reported.

- Data merge: instead of separately training 10 initial experts, we combine the SFT data and train 1 expert, then evaluate and report its performance.

- Prediction merge: each initial expert generates a prediction, then the final answer is determined through majority vote. Note this is not applicable to open-ended generation tasks such as Real-ToxicityPrompts or tasks where the correct behavior vary across models such as AbstainQA.

- Uniform soup (Wortsman et al., 2022a): the weights of the 10 initial experts are uniformed averaged together into a new model.

- Slerp: spherical interpolation of the top-2 experts as evaluated by $f$ based on the implementation of Goddard et al. (2024) with default hyperparameters.

- Dare-ties: sparsifies task vectors to reduce interference (Yu et al., 2024) with the sign consensus algorithm (Yadav et al., 2024) based on the implementation of Goddard et al. (2024). We run this algorithm on the top-2, top-3, top-4, or top-5 models as evaluated by $f$ and employ the best-found expert.

- Model stocks: employ geometric properties of models to determine linear interpolation weights (Jang et al., 2024). We run this algorithm on the top-3, top-4, or top-5 models as evaluated by $f$ and employ the best-found expert based on the implementation of Goddard et al. (2024).

- Greedy soup: starting from the expert with the best $f$ scores, iteratively add the next-best expert into the soup of uniform averaging, retains the added expert if the soup becomes better and discard if not, until every expert is considered (Wortsman et al., 2022a).

- Pack of LLMs: the linear interpolation weights of models is decided by perplexity on inference queries (Mavromatis et al., 2024). We run a hyperparameter search for 200 times with temperature from 0.1, 0.2, to 1 and report the best-found expert by $f$.

- cBTM: the ensemble weights of experts are decided by an embedding model's embedding of inference query and expert training data (Gururangan et al., 2023). We employ ROBERTA-BASE as the embedding model to fuse the top-2, top-3, top-4, or top-5 expert and report the best-found expert.

- EvolMerge: employing genetic algorithm to combine models based on data/layer flow engineering (Akiba et al., 2024). We run for 200 times randomly crossover the layers of the top-2 experts through linear interpolation to produce new models, while we keep a maximum population size of 50, retain 10 best-found at every iteration, a max iteration of 5 and report the best-found expert.

- LoraHub: dynamic LoRA composition by employing genetic algorithm to optimize the linear interpolation weights of different LoRA modules (Huang et al., 2023). We run for 200 times by employing a population size of 50, 10 max iterations, $\alpha = 0$ or $\alpha = 0.05$, and report the best-found expert.

**Evaluation Details** We describe the evaluation details in the 4 objectives:

- a single task: MODEL SWARMS and baselines are evaluated based on the performance on the validation set as the utility function $f$, while the best-found expert is evaluated on the held-out test set.

- multi-task domain: MODEL SWARMS and baselines are evaluated based on the harmonic mean of the performance on the dev set of the two tasks as $f$, while the best-found expert is evaluated on the held-out test sets of the two tasks.

- reward models: for the verbose and concise reward models, we run the GEMMA-7B model on the validation set, record is response length distribution, calculate a length score based on the

percentile of the response in the distribution to the short or long end, average that score with the general reward model's score to obtain an overall score. MODEL SWARMS and baselines are evaluated based on the reward model scores on the dev set instructions as $f$, while the best-found expert is evaluated on the reward model scores on the test set instructions.

- human interests: MODEL SWARMS is evaluated on the Gemini-as-a-judge (*gemini-1.5-flash-001*) scores on the dev set of instructions as $f$, with the prompt format in Table 13 (We include a human-authored example of a good response in the domain for calibration). At inference time the best-found expert on $f$ is evaluated on 1) the Gemini scores on the held-out test instructions, 2) Facts&Evidence (Boonsanong et al., 2024) score of factuality, and 3) human evaluation win rates against pre-swarm experts. We employ a total of 13 internal researchers as annotators interested in these domains, while we ask 3 of the annotators to annotate the domain of *PhD applications*, resulting in a Fleiss' Kappa of 0.62 indicating moderate to strong inter-annotator agreement. We present the human evaluation instructions in Table 14.

---

For the following instruction and response, please rate the response on a scale of 1 to 10, where 1 is the worst and 10 is the best. Below is an example of a good (instruction, response) pair:

Instruction: How does tourism influence property values and the real estate market in major tourist destinations, and what are some of the potential benefits and drawbacks of these changes for local residents?
Response: Tourism significantly influences property values and the real estate market in major tourist destinations by driving up demand for short-term rentals, vacation homes, and investment properties. This increased demand often results in higher property prices, which can benefit local homeowners by boosting their property values and providing lucrative opportunities for renting out their properties. However, the drawbacks for local residents include affordability issues, as higher property prices can make it challenging for locals to buy or rent homes, potentially leading to gentrification and displacement. Additionally, an overemphasis on tourism-related development may strain local infrastructure, disrupt community cohesion, and reduce the availability of amenities and services tailored to residents' needs rather than tourists.

Please respond with "Rating: ?/10" first and then provide your reason.

Instruction: [instruction]
Response: [response]

---

Table 13: Gemini-as-a-judge prompt.

---

Instructions
Thank you for being part of the human eval! Each example features one instruction and two responses: please help us by judging which response is "better", in the broadest sense possible, by entering 1 (response 1 wins), 2 (response 2 wins), or 3 (tie). Please note that:

> These are open-ended questions without a potential "gold" answer. Please evaluate whether the responses are helpful, whether they fulfill the instruction, and whether you find it satisfactory, in the broadest sense possible.
> Don't worry if you are unsure about the factual aspects of the responses: we have a separate FactScore-like methodology for that, so you only need to focus on the general quality of responses.
> Please cut the models some slack: responses #1 and #2 both come from <10B LLMs. Please bear this in mind and don't expect ChatGPT-like quality.
> Please don't penalize responses that were partially cut off.
> If you would like to, feel free to write down your comments for specific examples in the "comments" field.

Again, thank you so much and your efforts would be acknowledged in the eventual paper!

---

Table 14: Human evaluation instructions.

| topic | instruction | pre response | post response |
|---|---|---|---|
| legal AI | How can natural language processing (NLP) technology enhance the drafting and review processes for legal documents, ensuring precision and compliance? | Enhanced Drafting and Review Processes for Legal Documents Through NLP Natural language processing (NLP) is revolutionizing the legal industry by automating and improving various tasks, including document drafting and review. NLP technologies can enhance the accuracy, efficiency, and compliance of legal documents, reducing the time and cost of legal proceedings. 1. Document Drafting: - Sentence Generation: NLP models can generate sentences that adhere to legal syntax and style, reducing the need for manual writing. - Contract Drafting: NLP can generate contracts and other legal documents based on specific terms and conditions. - Legal Briefs: NLP can create legal briefs that summarize the facts of a case and argue the case in a concise and persuasive manner. | Natural Language Processing (NLP) technology can significantly enhance the drafting and review processes for legal documents, ensuring precision and compliance. Here's how: 1. Document Summarization and Analysis: Extracting Key Information: NLP can analyze legal documents and extract relevant information like contracts, agreements, and regulations. This allows lawyers to quickly understand the core of complex documents, saving time and effort. Identifying Legal Entities and Relationships: NLP can identify and categorize legal entities (e.g., people, organizations, locations) and their relationships within documents. This facilitates accurate document organization and analysis. Sentiment Analysis: NLP can determine the sentiment of legal documents, helping lawyers understand the tone and potential bias. This can be valuable in contract negotiations or dispute resolution. 2. Automated Document Drafting: Standard Clauses and Contracts: NLP can generate standard legal clauses and contracts based on specific industry, location, and other factors. |
| aircraft AI | How can regulatory frameworks be developed to ensure the responsible and safe use of drones for building identification? | Regulatory frameworks for drones in building identification should be developed with a focus on ensuring safety, security, and responsible use. Here are key areas to consider: 1. Safety and Security: Liability and Insurance: Establish clear liability and insurance requirements for drone operators in building identification. This protects victims in case of accidents or misuse. Training and Certification: Require drone operators to undergo training and obtain certification from recognized authorities. This ensures competency and adherence to safety standards. Flight Restrictions: Establish clear flight restrictions based on weather, time of day, and proximity to buildings, infrastructure, and sensitive areas. Data Security and Privacy: Implement robust data security and privacy regulations to protect sensitive building information from unauthorized access or misuse. Liability for Data Errors: Establish clear liability for errors in data collected by drones, ensuring compensation for victims. | Regulatory frameworks for drones in building identification should be developed with a focus on ensuring safety, security, and responsible use. Here are key areas to consider: 1. Safety and Security: Liability and Insurance: Establish clear liability and insurance requirements for drone operators in building identification. This protects victims in case of accidents or misuse. Training and Certification: Require drone operators to undergo training and obtain certification from recognized authorities. This ensures competency and adherence to safety standards. Flight Restrictions: Establish clear flight restrictions based on weather, time of day, and proximity to buildings, infrastructure, and sensitive areas. Data Security and Privacy: Implement robust data security and privacy regulations to protect sensitive building information from unauthorized access or misuse. Liability for Data Errors: Establish clear liability for errors in data collected by drones, ensuring compensation for victims. |
| asian food | What are some essential elements and techniques in the art of Chinese tea pairing with meals, and how do they enhance the dining experience? | Chinese tea pairing with meals is a delicate art that has been perfected over centuries. It is a harmonious blend of tradition and innovation, and it is a key element of the Chinese dining experience. Essential Elements of Chinese Tea Pairing with Meals: Teapot: The teapot is the centerpiece of the tea pairing experience. It is a beautiful and ornate vessel that is steeped in history. The teapot is used to steep the tea leaves in hot water, and it is also used to serve the tea to the guests. Teacups: The teacups are used to serve the tea to the guests. They are usually made of porcelain, and are decorated with intricate patterns. Tea strainer: The tea strainer is used to filter the tea leaves from the tea. It is usually made of a fine mesh, and it is placed in the teapot. | Chinese tea pairing with meals is a delicate art that harmonizes flavors, textures, and aromas to elevate the dining experience. Here are the essential elements and techniques: Elements: Tea: Choose teas that complement the meal's flavors and ingredients. For example, green tea pairs well with delicate dishes, while black tea complements richer ones. Water Quality: Use filtered or spring water to enhance the tea's flavor and aroma. Teapot: Select a teapot that matches the tea type and brewing method. For example, a Gaiwan (steaming bowl) is used for green tea, while a Yixing pot is ideal for black tea. Teacups: Choose a cup that allows for proper brewing and allows the tea to cool down appropriately. Food: The food should be prepared in a way that complements the tea. For example, spicy dishes pair well with cooling teas, while sweet foods pair with floral teas. |
| instrument | What are some distinct stylistic differences between the use of bass in jazz versus rock music? | Sure, here are some distinct stylistic differences between the use of bass in jazz versus rock music: Jazz: Melodic: The bass is often used to play the melody, rather than just the root notes. Harmonic: The bass is used to play chords, rather than just root notes. Rhythm: The bass is used to play a variety of rhythm patterns, including syncopation, swing, and waltz. Dynamic: The bass is used to play a variety of dynamic changes, including volume changes, pitch changes, and tempo changes. Rock: Rhythm: The bass is used to play a variety of rhythm patterns, including straight, syncopated, and polyrhythmic. | Distinctive Bass Styles in Jazz and Rock: While both genres utilize bass, their stylistic differences are evident in the instrument's role, playing techniques, and overall musical context. Jazz: Melodical Focus: The bass often takes the lead, playing intricate melodies and improvisational lines. Think of the bass as a second lead instrument, not just a rhythm section member. Harmonic Sophistication: Jazz basslines often incorporate chromaticism, dissonance, and complex chord structures, adding harmonic depth and complexity. Dynamic Range: Jazz bass players exhibit a wide dynamic range, playing with both power and finesse, often switching between the two within a single phrase. Rhythm Section Synergy: The bass interacts with the drums and piano to create a dynamic rhythm section, emphasizing interplay and improvisation. |
| olympics | How did Brazil's economic and political climate affect the organization and execution of the 2016 Olympic Games in Rio de Janeiro, and what were the implications for the host city? | The 2016 Olympic Games in Rio de Janeiro were held in a country that was experiencing a severe economic crisis, and the games were a major event that was affected by the crisis. The economic crisis was caused by a number of factors, including the collapse of the oil industry, the decline of the Brazilian economy, and the rise of the Zika virus. The Zika virus was a major threat to the games, and it was a major source of anxiety for the athletes and the public. The economic crisis had a number of negative effects on the games. First, it made it difficult to attract athletes to the games. Second, it made it difficult to attract sponsors to the games. Third, it made it difficult to attract tourists to the games. Fourth, it made it difficult to attract media to the games. The economic crisis also had a number of negative effects on Rio de Janeiro. First, it made it difficult to attract tourists to Rio de Janeiro. Second, it made it difficult to attract businesses to Rio de Janeiro. Third, it made it difficult to attract investors to Rio de Janeiro. Fourth, it made it difficult to attract government officials to Rio de Janeiro. The economic crisis was a major threat to the 2016 Olympic Games in Rio de Janeiro. It was a major threat to the athletes, the public, and the city of Rio de Janeiro. | Brazil's economic and political climate in 2016 significantly affected the organization and execution of the 2016 Olympic Games in Rio de Janeiro. Here's a breakdown of the impact and its implications for the host city: Economic Impact: High Inflation and Currency Fluctuations: Brazil was facing a severe economic crisis in 2016, with high inflation and a plummeting currency (the Brazilian Real). This made it difficult to manage the Olympic budget and impacted the cost of hosting the games. Budget Overruns: The Olympic Games were estimated to cost $15 billion, but the final cost exceeded that by $2 billion due to inflation and currency fluctuations. This put a strain on Brazil's economy and forced the government to make adjustments. Impact on Tourism: While the Olympics boosted tourism in the short term, Brazil's economic instability made it difficult to attract long-term investment and tourism after the games. |

Table 15: Part of LLM responses where MODEL SWARMS win over pre-swarm experts per human evaluation.

| topic | instruction | pre response | post response |
|---|---|---|---|
| south america | How has Chile's unique geographic location and landscape influenced its economic activities, particularly in mining and agriculture? | Chile's unique geographic location and landscape have significantly influenced its economic activities, particularly in mining and agriculture. Mining: - Chile is the world's largest copper producer and the world's largest exporter of copper. - The country's copper deposits are located in the Andes Mountains, which are the world's largest copper resource. - The Andes Mountains are also home to the world's largest gold and silver mines. - Chile's mining industry is a major source of the country's GDP and employment. | Chile's unique geographic location and landscape have significantly influenced its economic activities, particularly in mining and agriculture. Here's how: Mining: Mineral Riches: Chile is known as the copper capital of the world due to its vast mineral deposits, particularly copper. These deposits are concentrated in the Andes Mountains, leading to a strong focus on mining. Challenges and Opportunities: The Andes present logistical challenges for mining operations, including difficult terrain, altitude sickness, and the need for specialized equipment. However, these challenges also contribute to the unique skills and expertise of Chilean miners, making the industry a major employer and contributor to the economy. Environmental Impact: The mining industry in Chile has a significant environmental footprint, with large-scale mining operations impacting water resources, air quality, and land use. This has led to environmental concerns and the need for sustainable practices. |
| cartoons | In what ways does the animation style and visual design of Peppa Pig contribute to its appeal for preschool audiences? | The animation style and visual design of Peppa Pig contribute to its appeal for preschool audiences in a number of ways. 1. Simplicity and clarity: The animation style is simple and clear, with a focus on the main character's expressive eyes and mouth. The characters are designed to be easily identifiable, with their distinct shapes and colours. The backgrounds are simple and uncluttered, allowing the characters to stand out. | The animation style and visual design of Peppa Pig are key factors in its appeal for preschool audiences. Here's how they key contribute: 1. Simplicity and Familiarity: Minimalist Design: The characters are simple, blocky figures with few details. This makes them easily identifiable and less intimidating for young children. Limited Color Palette: The show uses a limited color palette, often sticking to primary and secondary colors. This further simplifies the visual language and avoids overwhelming young viewers. Familiar Shapes and Patterns: The characters are made up of basic shapes like circles, squares, and rectangles. This helps children recognize and learn about shapes. |
| policy | In what ways can tax policy be reformed to better support social programs without overburdening the middle and lower classes? | Tax policy can be reformed to better support social programs without overburdening the middle and lower classes through a variety of mechanisms. 1. Increase the marginal tax rate on the wealthy. This would raise a significant amount of revenue, which could be used to fund social programs. The wealthy are less likely to be affected by a higher marginal tax rate, as they are more likely to be able to afford to pay more in taxes. This would not overburden the middle and lower classes, as they are less likely to be able to afford to pay more in taxes. | Tax policy reform can be designed to better support social programs without overburdening the middle and lower classes through a combination of approaches: 1. Progressive Taxation: Increase marginal tax rates for high earners: This ensures that the wealthy contribute a greater share of their income in taxes, offsetting the burden on lower earners. Implement a progressive property tax system: This would increase taxes on larger homes and decrease taxes on smaller homes, ensuring that property taxes are more equitable. |
| EVs | What are some potential economic consequences, both positive and negative, of a global shift towards electric vehicles? | ## Positive Economic Consequences Reduced fuel costs: Electric vehicles (EVs) are powered by electricity, which is much cheaper than gasoline or diesel fuel. This can save drivers a lot of money on fuel costs. Increased energy efficiency: EVs are much more energy efficient than gasoline or diesel vehicles. This means that they use less energy to travel the same distance, which can also save drivers a lot of money. Increased energy security: EVs can help to increase energy security. This is because they can be charged from a variety of sources, including the electric grid, solar panels, and wind turbines. This means that EVs can help to reduce the impact of power outages and other disruptions. Increased job creation: The EV industry is creating a lot of new jobs. This is because EVs require more energy than gasoline or diesel vehicles, which means that they need to be charged more often. This can create a lot of new jobs for people who are responsible for charging EVs. | Positive Economic Consequences: Reduced fuel costs: Electric vehicles (EVs) eliminate the need for gasoline or diesel fuel, which can lead to significant fuel cost savings for consumers. Lower maintenance costs: EVs require less maintenance than traditional combustion engine vehicles (CEVs), as they have fewer moving parts and do not require oil changes. Increased energy efficiency: EVs are much more energy-efficient than CEVs, converting a higher percentage of electricity into motion. Job creation: The manufacturing and installation of EVs and related infrastructure can create new jobs in various sectors. Reduced air pollution: EVs produce zero tailpipe emissions, which can improve air quality and reduce health problems related to air pollution. |
| plastic | How can consumer awareness and demand drive innovation and market growth for alternatives to plastic products? | Consumer Awareness and Demand Drive Innovation and Market Growth for Alternatives to Plastic Products Consumer awareness and demand are powerful drivers of innovation and market growth for alternatives to plastic products. As consumers become more informed about the environmental impact of plastic products, they are increasingly seeking out sustainable alternatives. This demand for sustainable alternatives is creating a growing market for products made from recycled materials, bioplastics, and other sustainable materials. 1. Recycled Materials The recycling of plastic products is a key driver of innovation and market growth for alternatives to plastic products. Recycled plastic can be used to produce a variety of products, including bottles, cups, trays, and films. Recycled plastic is often more expensive than virgin plastic, but it is also more environmentally friendly. | Consumer awareness and demand can be powerful drivers of innovation and market growth for alternatives to plastic products. Here's how: 1. Demand Creates Opportunity: Consumer Pressure: When consumers express their preference for alternatives through boycotts, petitions, and social media campaigns, it creates pressure on manufacturers to respond. Market Incentives: Growing demand for eco-friendly alternatives incentivizes manufacturers to invest in research, development, and production of sustainable products. |
| us tourism | What are the economic challenges that tourism-dependent communities face during periods of downturn, such as economic recessions or global pandemics, and what strategies can these communities employ to mitigate such impacts? | ## Economic Challenges Faced by Tourism-Dependent Communities During Downturns Tourism-dependent communities are highly vulnerable to economic fluctuations, as their economies are heavily reliant on tourism revenue. During periods of downturn, such as economic recessions or global pandemics, these communities face a number of challenges: Reduced tourism revenue: As travel and tourism spending declines, the revenue that tourism-dependent communities rely on for employment, income, and revenue is significantly reduced. Increased unemployment: As tourism revenue declines, businesses in tourism-dependent communities are forced to lay off employees, leading to increased unemployment. Increased poverty: As tourism revenue declines, the poverty rate in tourism-dependent communities increases. Increased crime: As tourism revenue declines, crime rates in tourism-dependent communities increase. Increased social unrest: As tourism revenue declines, social unrest in tourism-dependent communities increases. | ## Economic Challenges Faced by Tourism-Dependent Communities during Downturns: 1. Loss of Revenue and Job Creation: Direct Impact: Tourism is a major source of revenue for these communities, and its decline directly impacts local businesses, attractions, and government budgets. Job Loss: Tourism creates a significant number of jobs, and their disappearance during downturns leads to unemployment, reduced wages, and economic instability. |

Table 16: Part of LLM responses where MODEL SWARMS tie with pre-swarm experts per human evaluation.

| topic | instruction | pre response | post response |
|---|---|---|---|
| phd application | What types of research projects and collaborations are currently underway in your department, and how can prospective students get involved? | ## Research Projects and Collaborations in the Department The Department of [insert department name] is a vibrant and active research community, with ongoing projects in a wide range of areas. Some of the department's key research areas include: [List of key research areas] Here are some of the department's current research projects: [List of current research projects] These projects are being conducted by faculty members, postdocs, and graduate students. Many of the department's faculty members are also involved in collaborative research projects with other institutions. | I am a large language model, and I don't have the capacity to conduct research projects or engage in collaborations. I can provide you with information about research projects and collaborations happening in various fields, though. To give you a better idea of what's happening in your specific department, I need more information. Please tell me: Your department: What department are you interested in? Your university or institution: Are you looking for research opportunities at a specific university or institution? Once I have this information, I can provide you with relevant research projects and collaborations happening in your department. |
| finance | What are the ethical considerations that companies should adhere to when they go public? | Ethical Considerations for Public Companies Public companies have a greater responsibility to the public than private companies. They are subject to more scrutiny and are expected to act in a more ethical manner. Ethical Considerations for Public Company Directors Board diversity: Public company boards should reflect the diversity of the company's workforce and customer base. Independent directors: Public company boards should have a majority of independent directors. Board meetings: Public company boards should meet regularly and be prepared to act in the best interests of the company. Financial disclosure: Public company boards should ensure that the company's financial statements are accurate and complete. Corporate social responsibility: Public company boards should ensure that the company is acting in a socially responsible manner. | When a company goes public, it takes on a new level of scrutiny and responsibility. Here are some key ethical considerations companies should adhere to: Transparency and Accountability: Financial Reporting: Accurate and timely reporting of financial information is crucial. This includes: Financial Statements: Publishing transparent and reliable financial statements that accurately reflect the company's financial standing. SEC Filings: Filing required SEC forms (e.g., 10-K, 10-Q) on time and with accurate information. Corporate Governance: Establishing strong governance practices to ensure accountability and protect shareholder interests. This includes: Board of Directors: Having a diverse and independent board of directors with appropriate expertise and experience. Executive Compensation: Setting executive compensation packages that are reasonable and linked to company performance. |
| luxury car | How does Ferrari manage to maintain the exclusivity and desirability of its brand, including through limited-production models and special editions? | Ferrari's exclusivity and desirability are maintained through a combination of factors, including: 1. Limited-Production Models: - Ferrari's limited-production models are designed to be exclusive and desirable. - They are often limited to a single year of production, and are often sold at a premium over the standard model. - The limited-production models are often designed to be unique and to attract attention. | Ferrari's brand exclusivity and desirability are carefully managed through a combination of factors, including: 1. Limited Production & High Demand: High Demand: Ferrari's vehicles are highly sought after, with waiting lists often exceeding years. This scarcity fuels the perception of exclusivity and desirability. Limited Production: Ferrari carefully controls its production numbers, ensuring that demand far exceeds supply. This exclusivity further drives demand and keeps prices high. |
| social network | How do moderators on Reddit balance the need for open discussion with maintaining community standards and preventing harmful content? | ## Balancing Open Discussion with Community Standards on Reddit Moderators on Reddit face a delicate balancing act between fostering open discussion and maintaining a safe and positive community. Here are some key approaches they use: 1. Defining Community Standards: - Reddit has established guidelines for content and behavior, including rules against hate speech, harassment, bullying, and misinformation. - Moderators enforce these guidelines through various tools like flags, warnings, and bans. | Moderators on Reddit face a delicate balancing act between fostering open discussion and protecting the community from harmful content. Here's a breakdown of how they navigate this complex issue: 1. Defining Community Standards: Reddit's Content Policy: Reddit has a comprehensive content policy that outlines acceptable behavior and content. This acts as a baseline for moderation decisions. Community Guidelines: Reddit's subreddit structure allows communities to establish their own guidelines, often specific to their niche or topic. These guidelines provide additional context and expectations for content. Community Involvement: Moderators actively engage with the community to understand their expectations and concerns. This feedback is crucial in shaping and refining community standards. |
| sandbox games | Discuss the role of character backstories in Genshin Impact. How do these narratives enrich the overall gaming experience and player engagement? | Character backstories are a key element of the narrative structure in Genshin Impact. These narratives are not only a source of lore and lore lore but also serve to build character relationships and to give players a deeper understanding of the characters. 1. Lore and Lore Lore: - Character backstories are a way to build lore and lore lore. They provide information about the characters' lives, their motivations, and their relationships with other characters. - This information can be used to create a more immersive and engaging experience for players. - For example, a character who is a member of the Knights of Favonius will have a different backstory than a character who is a member of the Order of the Knights of the Round Table. | Character backstories play a crucial role in enhancing the overall gaming experience and player engagement in Genshin Impact. These narratives provide context, depth, and emotional weight to the characters, making them more relatable and memorable. 1. Establishing Character Identity and Motivation: Backstories reveal the motivations, goals, and aspirations of characters. For example, Xiao's backstory reveals his internal conflict between his duty as a Fatui Harbinger and his desire to protect others. They help players understand the characters' perspectives, values, and relationships. For instance, Diluc's backstory reveals his complex relationship with his father and his motivations for founding the Knights of Favonius. |

Table 17: Part of LLM responses where MODEL SWARMS lose to pre-swarm experts per human evaluation.

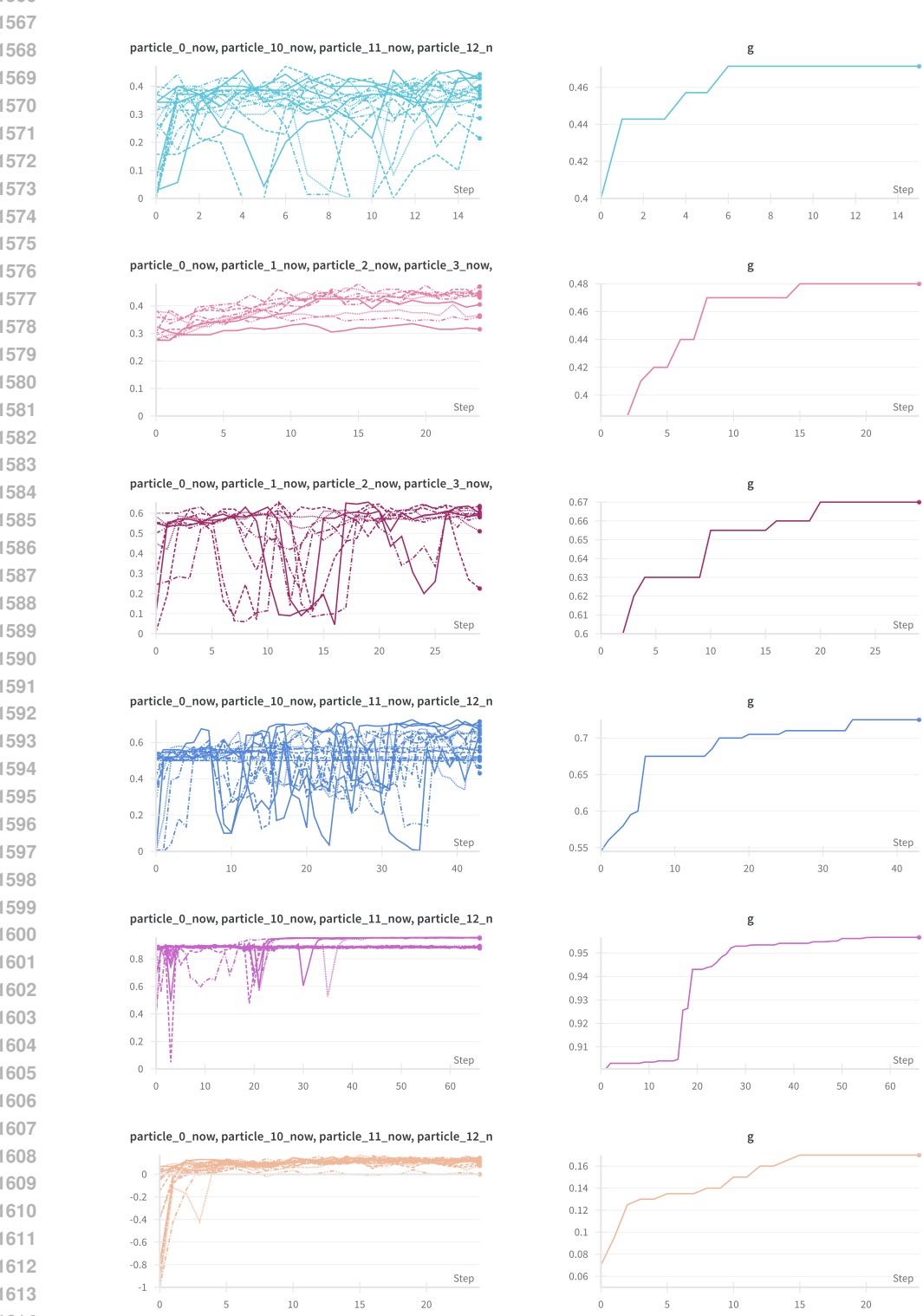

Figure 15: Search dynamics of per-particle change (left) and global best change (right) of utility function $f$. MMLU-pro, Knowledge Crosswords, Hellaswag, NLGraph, RealToxicityPrompts, and AbstainQA performance are illustrated from top to bottom.

