# OpenReview forum: "Model Swarms: Collaborative Search to Adapt LLM Experts via Swarm Intelligence"
_ICLR.cc/2025/Conference — Submitted to ICLR 2025_

### Official Review · Reviewer_bBJu · 2024-10-29

**Soundness:** 4
**Presentation:** 4
**Contribution:** 4
**Rating:** 8
**Confidence:** 4

**Summary:**

This paper explores swarm intelligence to guide LLM weight updates of a set of expert LLMs. In this work, a given set of LLMs are distributed in "weight-space" with randomized velocity and location (inspired by particle swarms). The algorithm updates the LLM particles such that local and global (collective) locations are considered to find new optimal locations, that is, model weights. Optimal locations are evaluated by benchmarking across various tasks from literature, as well as model composition/ensembling methods as baselines.

**Strengths:**

- Novel way of utilizing swarm intelligence (particles) to update weights of a given set of LLMs (experts)
- Very well presented in terms of language, explanations, and figures, but also evidence of why and how this works
- Extensive experimentation and very well-suited benchmarks/baselines
- Especially interesting experiments on the emergence of new skills, "Correctness Emergence", as well as the diversity of LLM sets ("Diversity Matters")

**Weaknesses:**

- I could not find evidence on how weights are mapped to the weight/location space. Please provide detailed explanation of how model weights are represented in the weight/location space, including any techniques used for dimensionality reduction or embedding if applicable.
- In line 110, there is a first mention of a utility function, but at this point, I think this should have been explained already. I would suggest to introduce and explain the utility function earlier in the paper, as it's a key concept for understanding the method. This would improve the paper's overall clarity and flow.
- line 154, I think it would be important to explain how to prevent experts that are best (globally) from being drawn to their own "best" initial location and how the random factor helps to explore. Please elaborate on how this issues is addressed, specifically detailing how the random factors in your method help prevent getting stuck in local optima and encourage exploration.
- line 258 - it might make sense to bring in SOTA numbers from non-composition LLMs
- line 318 - "in LLM-as-a-judge"
Here, I think we need more details on the prompts, etc. as there is evidence that LLM as a judge evaluation methods can be gamed easily. Could you provide more detail about the prompt and evaluation process? Perhaps discussing how you mitigate potential biases or gaming of this evaluation method.
- line 367
"Averaged across the four datasets, we found that only 10.4% of the ending-best particles also started as the best (#1), while surprisingly, the bottom half of the starting experts were able to rise to the top in 56.9% of the MODEL SWARMS searches."
This would be a good opportunity to further discuss the importance of the "global" values in the utility function, especially in the early iterations.

**Questions:**

- Paragraph following line 67 -  This should be explained at a high level at this point:
What does velocity map to?
What does "best found location" map to?
Why is it important to be in a "best" neighbourhood, and what does "best" refer to?
I suggest providing a high-level explanation of these concepts early in the paper, as this would help readers better understand the method.
- List following line 77
Is this the same model pool? Or for each point in the list of different pools?
- line 104
"location represented by model weights;" - Why are we not talking specifically about the weights being in an embedding space (locations) -  assuming this is the case? Otherwise, one would wonder how the weights can purely be taken from different size/architecture type models and mapped to the same space.
- line 171
"Since MODEL SWARMS explicitly encourage randomness and exploration," - how? This could be picked up briefly, here again, to circle back. "repeating" this here solidifies the idea

---

> ### Author Response · Authors · 2024-11-14
> **Author Response (1/2)**
>
> We would like to thank the reviewer for their thoughtful comments and feedback.
>
> > I could not find evidence on how weights are mapped to the weight/location space. Please provide detailed explanation of how model weights are represented in the weight/location space, including any techniques used for dimensionality reduction or embedding if applicable.
>
> > line 104 "location represented by model weights;" - Why are we not talking specifically about the weights being in an embedding space (locations) - assuming this is the case? Otherwise, one would wonder how the weights can purely be taken from different size/architecture type models and mapped to the same space.
>
> The location space is just the weight space itself, i.e. the location of a model is the parameter values of the model in the 7B-dimensional space. There is no dimensionality reduction or embedding.
>
> Since models only move towards the velocity direction, which is in turn impacted by the weight of other models, this is not a "free" search rather a constrained search of model compositions, so directly operating on the 7B model weight space is effective.
>
> In this sense, weights from different size/architecture could not be mapped to the same space. That’s why we have “Different Model Architectures with Token Swarms” and Figure 5 in Section 5, where we move the definition of the space to the token probabilities space, where token probability distributions are composed on-the-fly instead of weights when there are heterogeneous model architectures, inspired by proxy tuning [1]. Empirically we see that 4 Mistral-7B experts and 4 Gemma-7B experts could compose in the token probability space with the token swarms variant in Figure 5 and collectively become better.
>
> Embedding space is definitely another very promising space to consider when composing heterogeneous models. We are potentially exploring a follow-up on model heterogeneity and would be happy to explore alternative space definitions.
>
> [1] Liu, Alisa, et al. "Tuning language models by proxy." COLM 2024.
>
> > In line 110, there is a first mention of a utility function, but at this point, I think this should have been explained already. I would suggest to introduce and explain the utility function earlier in the paper, as it's a key concept for understanding the method. This would improve the paper's overall clarity and flow.
>
> We did introduce the utility function earlier in line 98:
>
> “It also requires a \emph{utility function} $f:\mathbf{x} \rightarrow \mathcal{R}$, mapping each expert onto a scalar value that should be optimized for model adaptation.”
>
> Let us know if you are suggesting something else.
>
> > line 154, I think it would be important to explain how to prevent experts that are best (globally) from being drawn to their own "best" initial location and how the random factor helps to explore. Please elaborate on how this issues is addressed, specifically detailing how the random factors in your method help prevent getting stuck in local optima and encourage exploration.
>
> > line 171 "Since MODEL SWARMS explicitly encourage randomness and exploration," - how? This could be picked up briefly, here again, to circle back. "repeating" this here solidifies the idea
>
> The global best expert is not drawn to their own “best initial location”. So when an expert is the global best, $p_i-x_i=0$ since the personal best is its current location $p_i=x_i$, similarly $g-x_i=0$ since the global best is also just its current location $g=x_i$. So the personal/global best terms have no effect on the current global best expert.
>
> About the effect of randomness: let’s assume there is one expert that is in a very high-utility location but not the highest. If there is no randomness, then the $\phi_g(g-x_i)$ would be very large and all models are drawn towards it very hard, leading to them stuck in the local optima. However, if there is a randomness factor $r_g \sim \mathcal{U}(0,1)$ that discounts the term as $r_g\phi_g(g-x_i)$ and $r_g$ happens to be very small, then it would allow the model to be less impacted by the global best term but more impacted by other terms to explore and move out of the local optima.
>
> We add to line 171 “this exploration is made possible by randomness factors $r_v, r_p, r_g, r_w \sim \mathcal{U}(0,1)$, where the impact of personal/global bests are randomly discounted to favor exploration rather than overly quick convergence."
>
> > line 258 - it might make sense to bring in SOTA numbers from non-composition LLMs
>
> The first two lines/baselines are non-composition LLMs. “Best Single” is the best-performing LLM expert in the pool decided by performance on the validation set. “Data Merge” is having one LLM trained on all SFT data that produced the pool of experts.
>
> > List following line 77 Is this the same model pool? Or for each point in the list of different pools?
>
> All evaluation types and tasks share the same model pool, described in Section 3.

---

> ### Author Response · Authors · 2024-11-14
> **Author Response (2/2)**
>
> > line 318 - "in LLM-as-a-judge" Here, I think we need more details on the prompts, etc. as there is evidence that LLM as a judge evaluation methods can be gamed easily. Could you provide more detail about the prompt and evaluation process? Perhaps discussing how you mitigate potential biases or gaming of this evaluation method.
>
> Table 11 presents the prompt for the LLM-as-a-judge evaluation. Yes automatic evals could be gamed, we take the following measure to enhance reliability: As you can see in Table 11, there is an (instruction, response) pair. The response is curated by querying PerplexityAI and then manually verified and edited. This would provide a high-quality instance for each domain for LLM grounding. We did some in-house quick eval to show that this is more stable than evaluation without a "gold" example.
>
> Table 4 also shows that this LLM eval is consistent with the conclusion of the human eval, that Model Swarms is consistently outperforming pre-swarm experts on both metrics.
>
> > line 367 "Averaged across the four datasets, we found that only 10.4% of the ending-best particles also started as the best (#1), while surprisingly, the bottom half of the starting experts were able to rise to the top in 56.9% of the MODEL SWARMS searches." This would be a good opportunity to further discuss the importance of the "global" values in the utility function, especially in the early iterations.
>
> We are adding the following sentence to line 372:
>
> “This indicates that the global best status is switching between experts frequently, suggesting that models are vibrantly and collectively improving and the top spot is constantly overtaken.”
>
> > Paragraph following line 67 - This should be explained at a high level at this point: What does velocity map to? What does "best found location" map to? Why is it important to be in a "best" neighbourhood, and what does "best" refer to? I suggest providing a high-level explanation of these concepts early in the paper, as this would help readers better understand the method.
>
> So “best” means “highest score on the utility function”, most commonly “highest score on the validation set”. In this sense:
>
> “Velocity” is the aggregation of model parameter differences. Say model 1 has weight $w_1$, model 2 has weight $w_2$, the $w_1-w_2$ is a “model parameter difference”, and velocity is an aggregation of various such terms (lines 150). This velocity term then guides where the model should move towards next.
>
> “Best found location” is then the model weight that achieves the highest utility function score across all experts’ search history.
>
> The “best neighborhood” is then valuable since “the weight neighborhoods of good model checkpoints might be promising to explore” (lines 146-147, [1])
>
> We add “best/worst meaning the best/worst-performing expert on the utility function” towards the end of line 69.
>
> [2] Eilertsen, Gabriel, et al. "Classifying the classifier: dissecting the weight space of neural networks." ECAI 2020.

---

> > ### Comment · Reviewer_bBJu · 2024-11-24
> >
> > Thank you very much for clarifying my questions and concerns. You can upload a revision btw. This is very interesting work. All the best.

---

### Official Review · Reviewer_jiVf · 2024-10-30

**Soundness:** 4
**Presentation:** 4
**Contribution:** 3
**Rating:** 8
**Confidence:** 4

**Summary:**

The paper introduces MODEL SWARMS, a collaborative search algorithm for adapting Large Language Models through swarm intelligence. The approach involves a pool of LLM experts that collectively optimize a utility function by adjusting in the weight space, guided by checkpoints from the most successful models. MODEL SWARMS offers tuning-free model adaptation and is designed to operate effectively even with minimal data (as few as 200 examples). The method is evaluated across various domains, showing improvements of up to 21.0% over 12 model composition baselines.

**Strengths:**

The proposed method is well-founded and practical, with wide applicability across diverse adaptation scenarios, from single-task to multi-task learning, as well as reward modeling and aligning with varied human interests.

This work is likely to inspire new directions in LLM research, providing valuable insights into adaptive model composition.

The paper is well-organized and clear, with a logical flow and thorough experimental design, making the findings compelling and easy to follow.

The experiments are comprehensive and detailed, offering a robust evaluation across various tasks and contexts.

**Weaknesses:**

No major faults were identified.

**Questions:**

Can this approach extend to decision-making tasks? Decision-making often requires the learning of a return or value function, which might pose challenges for stability in the search process. It would be insightful to understand how MODEL SWARMS handles stability and reliability under these conditions.

Would larger LLMs make the search more difficult as the parameter size increases?

---

> ### Author Response · Authors · 2024-11-14
> **Author Response**
>
> We would like to thank the reviewer for their thoughtful comments and feedback.
>
> > Can this approach extend to decision-making tasks? Decision-making often requires the learning of a return or value function, which might pose challenges for stability in the search process. It would be insightful to understand how MODEL SWARMS handles stability and reliability under these conditions.
>
> Absolutely yes, as long as the “return or value function” outputs a scalar value representing the utility of the decision. I wonder if the reviewer might have any pointers to decision making/planning data that could employed as evaluation for future work.
>
> > Would larger LLMs make the search more difficult as the parameter size increases?
>
> Conceptually no, as the particles/experts in a Model Swarm could be of any size, and the methodology isn’t impacted by size.
>
> Computationally the cost would certainly go up, linear to the amount of parameters, since the main computation cost comes from running model inference on a small validation set.
>
> Empirically it would be interesting to see whether model size has an impact on effectiveness: we by default employed Gemma-7B and we now quickly evaluate Gemma-2B experts on NLGraph, we see that:
>
> |    NLGraph    |       dev      |      test      |
> |:-------------:|:--------------:|:--------------:|
> |  pre-swarm 2B |      0.315     |      0.330     |
> | post-swarm 2B | 0.425 (+34.9%) | 0.420 (+27.2%) |
> |  pre-swarm 7B |      0.540     |      0.535     |
> | post-swarm 7B |  0.730 (+37.0%)  | 0.672 (+25.6%) |
>
> The improvement is consistently 20%-30% across 2B and 7B. We will try out models on a scale of 13B/70B when more compute is available.

---

> > ### Comment · Reviewer_jiVf · 2024-11-18
> >
> > Thanks for the response.

---

### Official Review · Reviewer_rBAc · 2024-11-02

**Soundness:** 3
**Presentation:** 3
**Contribution:** 2
**Rating:** 6
**Confidence:** 3

**Summary:**

The paper presents MODEL SWARMS, a collaborative search algorithm that adapts various Large Language Models (LLMs) for multiple applications. This method employs the best checkpoints as guides to optimize utility functions for different objectives, utilizing the concept of evolutionary games. Extensive testing demonstrated that MODEL SWARMS enhances performance by up to 21.0% compared to twelve conventional model composition baselines in four adaptation scenarios. Notably, this model does not require fine-tuning or presuppose the existence of expert models.

**Strengths:**

1. The concept is straightforward and the writing is clear and easy to comprehend.
2. This model operates without the need for fine-tuning and does not depend on already established expert models.
3. The performance appears to be quite strong relative to existing approaches.

**Weaknesses:**

1. Why did you choose Particle Swarm Optimization (PSO) for your model instead of other evolutionary algorithms such as Ant Colony Optimization (ACO) or Genetic Algorithms? Could you discuss the benefits of using PSO compared to these other common evolutionary game theory (EGT) strategies in your context?

2. Evolutionary Game Theory (EGT) is often criticized for its lengthy and unstable search times, as well as the significant computational resources required. Could you provide details on the computational time associated with your design?

**Questions:**

see Weaknesses

---

> ### Author Response · Authors · 2024-11-14
> **Author Response**
>
> We would like to thank the reviewer for their thoughtful comments and feedback.
>
> > Why did you choose Particle Swarm Optimization (PSO) for your model instead of other evolutionary algorithms such as Ant Colony Optimization (ACO) or Genetic Algorithms? Could you discuss the benefits of using PSO compared to these other common evolutionary game theory (EGT) strategies in your context?
>
> Lines 514-529 discuss why PSO is uniquely fitting for composing and adapting LLM experts. To recap:
>
> Genetic algorithm requires much more manual engineering. For example, several crossover operations and several mutation operations are required to produce ``offsprings’’ from existing entities, in this case, LLMs. While there are preliminary attempts at designing these operations, they rely on heuristics and manual engineering that are not scalable or flexible, and they are mostly applicable to automated prompt design only.
>
> For Ant Colony, the problem needs to be transformed into the format of finding the shortest paths on a weighted network. It is unclear how composing LLM model parameters could become a graph problem and whether ACO is applicable in this case.
>
> That said, we agree that evolutionary LLM design is a promising direction and there are many classic algorithms to further explore beyond Model Swarms, an important first step to this direction.
>
> > Evolutionary Game Theory (EGT) is often criticized for its lengthy and unstable search times, as well as the significant computational resources required. Could you provide details on the computational time associated with your design?
>
> For stability, Table 7 shows the performance on dev/test sets where Model Swarms is repeated for multiple runs. Despite the randomness, Model Swarms consistently find expert models better than any baseline in 73% of the runs, indicating stable improvements.
>
> For computational time, we refer the reviewer to lines 1160-1183 in the appendix. To recap, Model Swarms has linear complexity to the number of models and to the cost of one model inference, while it takes about 10-15 iterations for each run on average. Empirically, with 5 40GB GPUs you could run a Model Swarms search under an hour, with 2 GPUs you need about 3 hours. If that is still not fast enough, we propose further acceleration with dropout-k and dropout-n: we refer the reviewer to lines 483-488 and Figure 9.

---

> > ### Comment · Reviewer_rBAc · 2024-11-17
> >
> > Thank you for your reply. I will update my rating to 6.

---

> ### Author Response · Authors · 2024-11-17
>
> Thank you for raising the score! I wonder if the reviewer has changed the score: it is still appearing as 5 on our end, but maybe this is just a lag in website update.
>
> Thank you,
> Authors

---

> > ### Comment · Reviewer_rBAc · 2024-11-17
> >
> > I don't believe it's possible to adjust the rankings at this point. The final recommendation is scheduled for submission by November 26th.

---

> ### Author Response · Authors · 2024-11-17
>
> Reviewers should be able to change the score any time. The reviewer could try this: go to the top of your review, there’s an “edit” button on the top right, click it and change the score, then save.

---

> > ### Comment · Reviewer_rBAc · 2024-11-17
> >
> > Done, thanks:)

---

### Official Review · Reviewer_cETM · 2024-11-03

**Soundness:** 3
**Presentation:** 3
**Contribution:** 2
**Rating:** 5
**Confidence:** 4

**Summary:**

The paper proposes a collaborative search algorithm that uses swarm intelligence to adapt LLMs.  The method proposed starts with different LLM experts and a utility function; collaboratively the LLMs experts optimize the utility function. Each expert is a particle in the swarm. It has a location that depends on the model weights, a velocity which is the direction where it should move next, the personal best found location, and the global best and worst location over the entire history.  The population is initialized using crossover, the velocity is updated as a weighted average of the current velocity, personal best, global best, and global worst.  Weights and locations are updated in steps. The length of the steps is reduced over time until convergence. The results are compared with baselines for different ways of doing the composition (trivial, static, dynamic) on multiple datasets. The examples shown in the paper work in multiple cases, i.e., for a single task, multi-tasks, reward models, and different human interests.  The results overall are good.

**Strengths:**

The method proposed is training free and can be used with a small number of examples.

It is interesting to see how the collaboration of week models can outperform strong models.

The experimental results presented are overall positive, showing the method is a promising approach over a large variety of domains.

**Weaknesses:**

There are no examples in the paper of the kind of text the LLMs experts produce. There are detailed examples in the appendix but showing an example would have gone a long way in helping the readers understand the significance of the work.

The method proposed is a variation on Particle Swarm Optimization and Genetic Algorithms, so the innovation seem limited to apply those methods to LLMs.

The paper is not an easy read. The tables are dense, there are a lot of charts, but since no examples have been included, they feel too abstract.

**Questions:**

Can you explain what changes done compared to the PSO and GA algorithms are the most useful to achieve the results obtained?

---

> ### Author Response · Authors · 2024-11-14
> **Author Response (1/2)**
>
> We would like to thank the reviewer for their thoughtful comments and feedback.
>
> > There are no examples in the paper of the kind of text the LLMs experts produce. There are detailed examples in the appendix but showing an example would have gone a long way in helping the readers understand the significance of the work.
>
> > The paper is not an easy read. The tables are dense, there are a lot of charts, but since no examples have been included, they feel too abstract.
>
> We add examples of the LLM expert-generated texts to the main paper by appending a few lines in Table 4. Specifically:
>
> ```
> Q: How can natural language processing (NLP) technology enhance the drafting and review processes for legal documents, ensuring precision and compliance?
>
> Pre-swarm: Enhanced Drafting and Review Processes for Legal Documents Through NLP Natural language processing (NLP) is revolutionizing the legal industry by automating and improving various tasks, including document drafting and review. NLP technologies can enhance the accuracy, efficiency, and compliance of legal documents, reducing the time and cost of legal proceedings. NLP technology could generate many sentences, documents, and content that could be helpful to legal professionals.
>
> Post-swarm: Natural Language Processing (NLP) technology can significantly enhance the drafting and review processes for legal documents, ensuring precision and compliance. Here’s how: 1. Extracting Key Information: NLP can analyze legal documents and extract relevant information like contracts, agreements, and regulations. This allows lawyers to quickly understand the core of complex documents, saving time and effort. 2. Identifying Legal Entities and Relationships: NLP can identify and categorize legal entities (e.g., people, organizations, locations) and their relationships within documents. This facilitates accurate document organization and analysis. 3. Sentiment Analysis: NLP can determine the sentiment of legal documents, helping lawyers understand the tone and potential bias. This can be valuable in contract negotiations or dispute resolution.
> ```
>
> Quantitative evaluation for the above example shows that the post-swarm expert generates better (Gemini judge score of 5->8 out of 10) and more factual (Facts&Evidence score 0.478 -> 0.696) response in the legal AI domain. Another example:
>
> ```
> Q. What are some distinct stylistic differences between the use of bass in jazz versus rock music?
>
> Pre-swarm: Sure, here are some distinct stylistic differences between the use of bass in jazz versus rock music: The bass is often used to play the melody, rather than just the root notes. The bass is used to play chords, rather than just root notes. The bass is used to play a variety of rhythm patterns, including syncopation, swing, and waltz. The bass is used to play a variety of dynamic changes, including volume changes, pitch changes, and tempo changes.
>
> Post-swarm: While both genres utilize bass, their stylistic differences are evident in the instrument’s role, playing techniques, and overall musical context.
> Melodical Focus: The bass often takes the lead, playing intricate melodies and improvisational lines. Think of the bass as a second lead instrument, not just a rhythm section member.
> Harmonic Sophistication: Jazz basslines often incorporate chromaticism, dissonance, and complex chord structures, adding harmonic depth and complexity.
> Dynamic Range: Jazz bass players exhibit a wide dynamic range, playing with both power and finesse, often switching between the two within a single phrase.
> Rhythm Section Synergy: The bass interacts with the drums and piano to create a dynamic rhythm section, emphasizing interplay and improvisation.
> ```
>
> For the above example, we again see Gemini judge score improvement from 5 to 7 out of 10 and Facts & Evidence score improvement from 0.444 to 0.722.
>
> By appending a few lines in Table 4 to include these examples, we hope these expert generated texts would strengthen the main paper.

---

> > ### Comment · Reviewer_cETM · 2024-11-27
> >
> > Thanks, the addition of the examples helps a lot. I had seen them in the appendix, but not everyone reads the appendix. Unfortunately, I still find the paper hard to read.  Some tables are not in numerical order. They are dense and with limited explanations.  You have done a lot of work, but reading many tables full of numbers does not help understanding the work.
> > I will increase my score, but I think the paper needs a major rewriting.

---

> ### Author Response · Authors · 2024-11-14
> **Author Response (2/2)**
>
> > The method proposed is a variation on Particle Swarm Optimization and Genetic Algorithms, so the innovation seem limited to apply those methods to LLMs.
>
> > Can you explain what changes done compared to the PSO and GA algorithms are the most useful to achieve the results obtained?
>
> Model Swarms is not related to genetic algorithms (GA). It is related to Particle Swarm Optimization (PSO), another instance of evolutionary algorithms: PSO was proposed to solve classic optimization problems and is vastly different from Model Swarms. The key differences are:
>
> **1. Populate initial experts**. While PSO starting points could be randomly sampled from the weight space, it is simply impossible to randomly sample LLM weights due to the ultra-high dimensionality. Lines 136-140 describe Model Swarms’ unique way of growing the initial set of experts as starting points.
>
> **2. Global worst**. LLMs often have the weight continuity/smoothness that classic optimization problems don’t [1]: if you change a few parameters by 1e-3, the output would still be the same (but for non-convex non-continuous optimization problems, this is not the case). Given this unique attribute, we track the global worst expert and push the swarm away from bad model checkpoints (lines 114-117, lines 122-126).
>
> **3. Restarting failed experts**. While evaluating a particle on classic optimization problems is lightweight, evaluating an LLM expert on an objective is not trivial. To make this feasible and efficient, we propose to restart failed experts, experts who are not becoming better in $c_r$ iterations, by reverting it back to their personal best. This encourages diverse experts to freely explore and will restart them to have another chance in case of failure. (lines 171-176)
>
> Table 5, Figure 9, and Table 6 demonstrate the uniqueness of Model Swarms and the contribution of these bullet points.
>
> [1] Eilertsen, Gabriel, et al. "Classifying the classifier: dissecting the weight space of neural networks." ECAI 2020.

---

> > ### Comment · Reviewer_cETM · 2024-11-27
> >
> > Thanks for the explanations. I know that PSO is different from GAs, but your writing is confusing. In the paper, you mentioned GAs, when you said (lines 514-517): "MODEL SWARMS is in part inspired by particle swarm optimization (PSO) (Kennedy & Eberhart, 1995), an evolutionary algorithm (EA) solving optimization problems. This echoes a recent and contemporary uptake of EAs, especially genetic algorithms (GAs) in ML/LLMs ..."

---

> ### Author Response · Authors · 2024-11-25
> **Revised Version**
>
> We are thankful for your constructive comments and feedback: we have incorporated all your suggested edits and posted an updated version. **Readers can now find examples of generated texts in Table 5 on page 6 in the main paper.** In addition, the updates include but are not limited to presenting newly added experiments and results, clarifying methodology and experiment design, and providing more analysis and discussion on results and future work. We would appreciate it if you might have any further feedback.
>
> Thank you, authors

---

> ### Author Response · Authors · 2024-11-27
>
> Thank you for engaging and raising the score.
>
> > Thanks for the explanations. I know that PSO is different from GAs, but your writing is confusing. In the paper, you mentioned GAs, when you said (lines 514-517): "MODEL SWARMS is in part inspired by particle swarm optimization (PSO) (Kennedy & Eberhart, 1995), an evolutionary algorithm (EA) solving optimization problems. This echoes a recent and contemporary uptake of EAs, especially genetic algorithms (GAs) in ML/LLMs ..."
>
> "We mentioned GA" doesn't mean that "Model Swarms is based on GA". The quoted sentences basically say: Model Swarms is inspired by PSO, PSO is one example of EA, EAs are now explored in LLMs, with most research using GA in the EA family.
>
> > Thanks, the addition of the examples helps a lot. I had seen them in the appendix, but not everyone reads the appendix. Unfortunately, I still find the paper hard to read. Some tables are not in numerical order. They are dense and with limited explanations. You have done a lot of work, but reading many tables full of numbers does not help understanding the work. I will increase my score, but I think the paper needs a major rewriting.
>
> We uploaded a revised version where the tables are in numerical order. We believe in the benefit of robust and extensive evaluation, this is why we provided so many experimental results in the paper.

---

### Public Comment · ~Bedionita_Soro1 · 2024-11-26

Thank you for sharing your paper. After reviewing it, I have a few questions and suggestions:

1. Could you clarify what you mean by "tuning-free" in your approach? I noticed that you use particle swarm optimization, which seems to involve parameter tuning (reference lines 15-16).

2. Do you plan to release the source code for your implementation? This would be valuable for reproducibility and further research.
3. do you use a particular framework like lm-eval for training the experts

3. Regarding the paper length: The submission guidelines specify a 10-page limit, but your manuscript is 11 pages. Unless the guidelines have changed, you may want to consider adjusting the content to meet the page limit.

Best regards

---

> ### Author Response · Authors · 2024-11-26
>
> Thank you for your comment!
>
> 1. By "tuning-free" we mean that the change to model weights are not based on fine-tuning: there's no loss, back propagation, and backward pass. This is compared to learn-to-fuse approaches in existing works (lines 37-40). Model Swarms only conducts forward pass of model inference for model composition.
>
> 2. Yes, with the final version.
>
> 3. We used the HuggingFace SFTTrainer.
>
> 4. The submission ended right on the 10th page. Ethics limitations and reproducibility statements are permitted and encouraged to appear on the 11th page according to the CfP, as are many other submissions.

---

### Meta-Review · Area_Chair_AiQj · 2024-12-19

**Metareview:**

This paper seems to explore an interesting concept but the main concern here is that its technical narrative is not clear. There is very little explanation of the main algorithm.

For example, in lines 122-123, there is a non-trivial form for the velocity update that is hard to parse. I do notice that there is a bit of explanation regarding its terms but it remains unclear why these terms should be put together as such. Likewise, the reasoning for the update in lines 125-126 is somewhat heuristic & does not provide a clear insight into its nature.

As such, despite the positive empirical studies, the paper seems to come across as a black-box contribution due to the lack of elaboration for the main algorithm. This is a difficult decision but I believe in the current state, the paper does not meet the acceptance bar as its empirical performance is not properly explained.

**Additional Comments On Reviewer Discussion:**

There has been some discussion among the authors and reviewers. Although this paper has a set of mostly positive scores, most of the positive reviews are extremely light (particularly, those of reviewers rBAc and jiVf).

As a matter of fact, I have voiced out particular concerns regarding several unclear parts of the main algorithm (almost 2 weeks ago) and explicitly ask for further thoughts but none of the positive reviewers are willing to provide further clarification.

Given this, I believe the positive ratings are somewhat inflated and do not reflect the true quality of this paper. This explains why my decision is not aligned with the majority here.

---

### Decision · Program_Chairs · 2025-01-22

Reject